# Digestive contents and food webs record the advent of dinosaur supremacy

Martin Qvarnström[1✉], Joel Vikberg Wernström[1,2], Zuzanna Wawrzyniak[3], Maria Barbacka[4,5], Grzegorz Pacyna[6], Artur Górecki[6], Jadwiga Ziaja[5], Agata Jarzynka[7], Krzysztof Owocki[8], Tomasz Sulej[8], Leszek Marynowski[3], Grzegorz Pieńkowski[9,10], Per E. Ahlberg[1] & Grzegorz Niedźwiedzki[1,9✉]

The early radiation of dinosaurs remains a complex and poorly understood evolutionary event[1–4]. Here we use hundreds of fossils with direct evidence of feeding to compare trophic dynamics across five vertebrate assemblages that record this event in the Triassic–Jurassic succession of the Polish Basin (central Europe). Bromalites, fossil digestive products, increase in size and diversity across the interval, indicating the emergence of larger dinosaur faunas with new feeding patterns. Well-preserved food residues and bromalite-taxon associations enable broad inferences of trophic interactions. Our results, integrated with climate and plant data, indicate a stepwise increase of dinosaur diversity and ecospace occupancy in the area. This involved (1) a replacement of non-dinosaur guild members by opportunistic and omnivorous dinosaur precursors, followed by (2) the emergence of insect and fish-eating theropods[5–7] and small omnivorous dinosaurs. Climate change in the latest Triassic[5–7] resulted in substantial vegetation changes that paved the way for ((3) and (4)) an expansion of herbivore ecospace and the replacement of pseudosuchian and therapsid herbivores by large sauropodomorphs and early ornithischians that ingested food of a broader range, even including burnt plants. Finally, (5) theropods rapidly evolved and developed enormous sizes in response to the appearance of the new herbivore guild. We suggest that the processes shown by the Polish data may explain global patterns, shedding new light on the environmentally governed emergence of dinosaur dominance and gigantism that endured until the end-Cretaceous mass extinction.

Dinosaurs evolved in the mid-part of the Triassic, as indicated by the earliest unequivocal dinosaur fossils in upper Carnian deposits[8] and the remains of close dinosaur ancestors in the Middle Triassic[9]. However, terrestrial ecosystems dominated by dinosaurs of various trophic levels and taxonomic affinities, a structuring that would persist until the end-Cretaceous mass extinction, did not appear until the Early Jurassic, some 30 million years later[10]. Many non-dinosaur tetrapods (for example, most temnospondyl amphibians, procolophonid pararreptiles, rhynchosaurs, phytosaurs and pseudosuchians, and some therapsids) became extinct during this interval, leading to the rise of dinosaurs being considered one of the most classic examples of a macroevolutionary biotic replacement. Two main contrasting models have been proposed to explain this event. The traditional 'competitive replacement model' argues that dinosaurs outcompeted their rivals because of more efficient physiologies, new anatomical adaptations or different feeding habits[11,12]. By contrast, the 'opportunistic replacement' model focuses on the role of stochastic processes that would have enabled the early radiation of dinosaurs following a diversity decline, or total extinction, of other groups[13–15]. There are still various opinions on the impact of the mass extinction at the end of the Triassic on the evolutionary success of dinosaurs[7,16]. New findings and more accurate chronostratigraphic dating have improved our understanding of the patterns of early Mesozoic tetrapod evolution[17]. However, no single hypothesis seems capable of explaining the rise of dinosaurs fully and critical questions about how dinosaurs established their dynasty on land remain largely unanswered[18–24].

The Late Triassic–earliest Jurassic tetrapod communities from the Polish Basin, the eastern sub-basin of the Central European Basin (Fig. 1 and Supplementary Information), represent snapshots of principal stages of early dinosaur evolution, representing assemblages in which (1) dinosaur precursors, dinosauriforms, had a minor ecological role (the Krasiejów–Woźniki biota, mid–late Carnian); (2) the first predatory dinosaurs began to diversify (the Poręba–Kocury biota, mid–late Norian); (3) early dinosaurs had a moderate ecological role (Lisowice–Marciszów biota, late Norian–earliest Rhaetian); (4) the first large herbivorous dinosaurs, sauropodomorphs, appeared (the Gromadzice–Rzuchów biota, mid–late Rhaetian); and (5) diversified saurischian and ornithischian dinosaurs completely dominated the terrestrial ecosystem (the Sołtyków–Hucisko biota, latest Rhaetian–earliest Hettangian).

[1]Department of Organismal Biology, Uppsala University, Uppsala, Sweden. [2]The Arctic University Museum of Norway, UiT The Arctic University of Norway, Tromsø, Norway. [3]Institute of Earth Sciences, Faculty of Natural Sciences, University of Silesia in Katowice, Sosnowiec, Poland. [4]Hungarian Natural History Museum, Botany Department, Budapest, Hungary. [5]W. Szafer Institute of Botany, Polish Academy of Sciences, Kraków, Poland. [6]Institute of Botany, Department of Taxonomy, Phytogeography and Palaeobotany, Faculty of Biology, Jagiellonian University, Kraków, Poland. [7]Institute of Geological Sciences, Polish Academy of Sciences, Research Centre in Kraków, Kraków, Poland. [8]Institute of Paleobiology, Polish Academy of Sciences, Warsaw, Poland. [9]Polish Geological Institute – National Research Institute, Warsaw, Poland. [10]Deceased: Grzegorz Pieńkowski. ✉e-mail: martin.qvarnstrom@ebc.uu.se; grzegorz.niedzwiedzki@ebc.uu.se

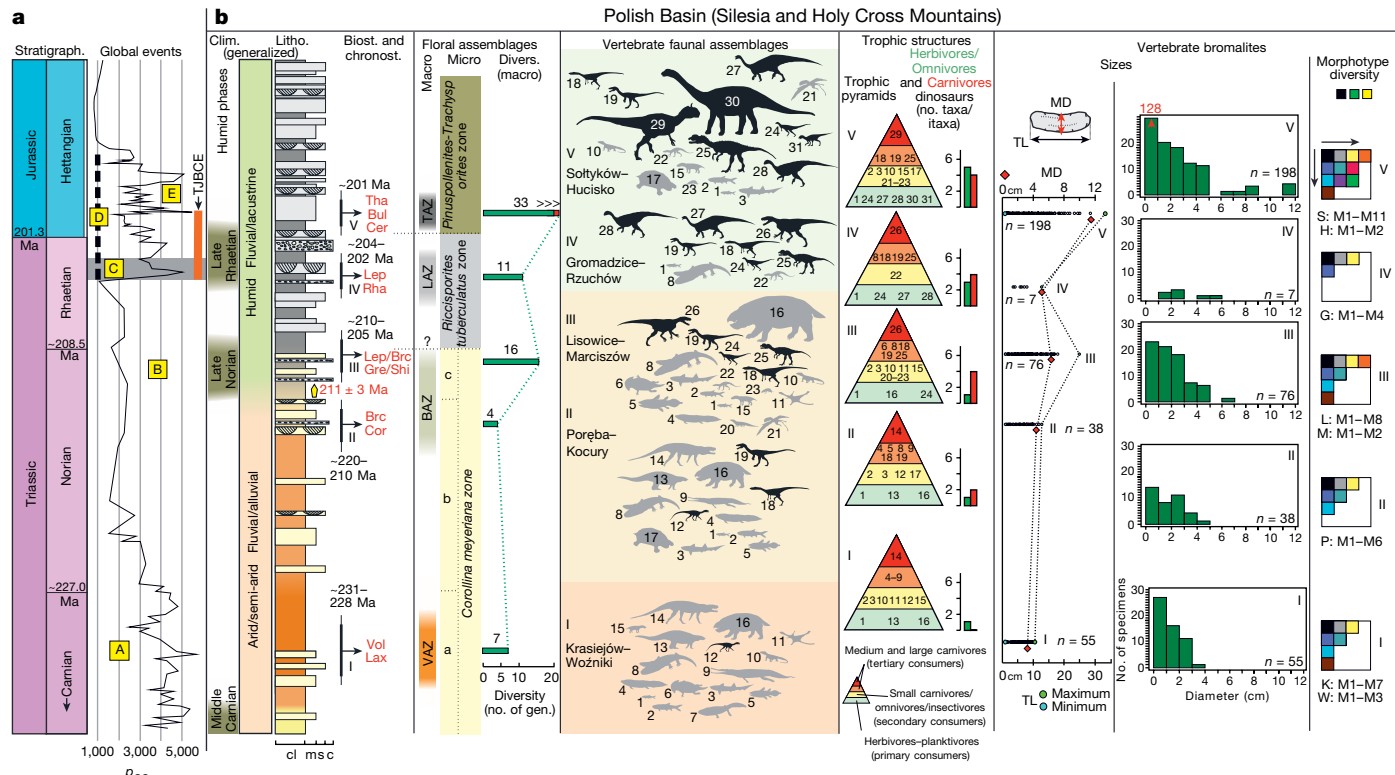

**Fig. 1 | Terrestrial vertebrate assemblages from the Polish Basin across the Carnian to Hettangian. a**, Global events: curve of partial pressure of $CO_2$ in ppm. A, oldest dinosaurs; B, the first phase of dinosaur radiation (late Norian); C, end-Triassic extinction; D, CAMP volcanism; E, the first ecosystems dominated by dinosaurs. **b**, Geological and fossil record from the Polish Basin. Clim., climatology; Litho., lithology; Biost., biostratigraphy; Chronost., chronostratigraphy; Divers., diversity; TJBCE, Triassic–Jurassic boundary climatic events in the Polish Basin[7]; gen., genera; Vol, *Voltzia* flora; Lax, *Laxitextella* conchostracan fauna; Cor, *Corollina meyeriana* palynoflora; Brc, *Brachyphyllum* flora; Gre/Shi, *Gregoriusella/Shipingia* conchostracan fauna; Lep/Brc, *Lepidopteris/Brachyphyllum* flora; Lep, *Lepidopteris* flora; Rha, *Rhaetipollis germanicus*; Cer, *Cerebropollenites thiergartii*; Bul, *Bulbilimnadia* conchostracan fauna; Tha, *Thaumatopteris* flora; VAZ, *Voltzia* floral assemblage zone; BAZ, *Brachyphyllum* floral assemblage zone; LAZ, *Lepidopteris* floral assemblage zone; TAZ, *Thaumatopteris* floral assemblage zone. Vertebrate diversity (Supplementary Table 2): I–V, vertebrate assemblages (dinosauriforms represented by black and non-dinosauriform vertebrates by grey silhouettes).

1, small actinopterygian fish; 2, large actinopterygian fish; 3, hybodont (Hybodontiformes) and rhomphaiodon (Synechodontiformes) sharks; 4, dipnoan fish; 5, coelacanth fish; 6, plagiosaurid temnospondyls; 7, trematosaurid temnospondyls; 8, capitosaurid temnospondyls; 9, phytosaurs; 10, lepidosauromorphs; 11, sharovipterygids; 12, silesaurids; 13, aetosaurs; 14, rauisuchians; 15, cynodonts; 16, dicynodonts; 17, turtles; 18, small basal theropods; 19, small neotheropods; 20, thalattosaurids; 21, pterosaurs; 22, crocodylomorphs; 23, mammaliaforms/mammals; 24, ornithischian dinosaurs; 25, large neotheropods; 26, large predatory dinosauriforms/basal theropods; 27, quadrupedal sauropodomorphs; 28, bipedal sauropodomorphs; 29, giant neotheropods; 30, large sauropodomorph/early sauropod; 31, ?heterodontosaurids. Itaxa, ichnotaxa. Trophic structures: the numbers used in the trophic pyramids correspond to those presented in the faunal assemblages. Vertebrate bromalites: TL, total length; MD, maximum diameter; S, Sołtyków; H, Hucisko; G, Gromadzice; L, Lisowice; M, Marciszów; P, Poręba; K, Krasiejów; W, Woźniki. The $p_{CO_2}$ curve is based on soil carbonate proxies[59]. For other and more detailed data from the Polish Basin, see the Supplementary Information.

Here we analyse the rise and early evolutionary radiation of dinosaurs using a completely new approach. We used an array of methods, including synchrotron microtomography (Methods and Supplementary Table 1), to perform analyses of more than 500 bromalites (coprolites, cololites and regurgitalites) and other fossils with direct evidence of feeding (for example, bones with signs of predation/scavenging). We used these data to reconstruct trophic structures in the five fossil assemblages from the terrestrial Late Triassic and earliest Jurassic record of Poland (Fig. 1, Supplementary Information and Supplementary Tables 2–9) to see how they evolved over time. In addition, a range of palaeoenvironmental data were compiled to investigate the timing of climate and environmental changes relative to floral and faunal turnovers[7,25–28] (Fig. 1). The fossil assemblages were selected to span the interval of interest in a well-correlated, restricted geographical area, which is important because the rise of dinosaur supremacy was probably diachronous across Pangaea (for example, as a result of climatic barriers spanning north–south)[18,24]. The most comprehensive data for detailed food web reconstructions derive from the three most fully described biotas (denoted as 1, 3 and 5 above and I, III and V in Fig. 1), known from thousands of skeletal and trace fossils. The other fossil assemblages complement our knowledge of the plant and vertebrate diversity in the area, and the ecosystem changes that occurred between them over time.

## Late Triassic food webs

The Late Triassic bromalites show a big disparity (Fig. 1, Supplementary Information and Supplementary Tables 3–9) and contain a wide range of food remains, including tetrapod bone and tooth fragments, fish remains (sometimes articulated), plants, bivalves and exceptionally preserved arthropods, including numerous beetles and a cycloid larva (Fig. 2, Extended Data Figs. 1–10 and Supplementary Information). The repeatability of their shapes and contents allows a categorization of morphological groups and identification of probable producers (Supplementary Information).

Bromalites that were subjected to molecular analysis (specimens from Poręba and Lisowice) share similar molecular compositions, with a prominence of the polar fraction and a small proportion of aromatic compounds (Supplementary Information). The preservation of labile organic compounds such as α- and β-amyrins, α-amyrone, sterols, palmitin, stearin and levoglucosan attests to rapid mineralization of the

faeces at very early stages of diagenesis. The identification of such compounds as *n*-alkanes with predominance of short-chain homologues supports the influence of bacteria during this mineralization process.

The mid–late Carnian Krasiejów–Woźniki vertebrate community (Fig. 1 and Supplementary Information), the oldest of the studied assemblages, was composed of fish (hybodont sharks, actinopterygians and sacropterygians), temnospondyls (*Cyclotosaurus*, *Metoposaurus* and a plagiosaurid), therapsids (the medium-sized dicynodont *Woznikella* and the small eucynodont *Polonodon*) and various archosauromorphs (*Ozimek, Paleorhinus, Polonosuchus, Stagonolepis*), including the omnivorous dinosauriform *Silesaurus* – the only known dinosaur relative in the biota[29]. Even if the precise time relation between the two fossil sites that make up this biota is not fully understood, finds of invertebrates (arthropods, bivalves) and certain tetrapods (a silesaurid and an eucynodont) indicate a very similar age[30,31].

The younger Poręba–Kocury community shows that dinosaur precursors persisted alongside true theropod dinosaurs into the mid–late Norian in the region (Fig. 1 and Supplementary Information). Silesaurids and early saurischians (a supposed herrerasaurid and a potential neotheropod) probably constituted a common ecological structuring during this stage of early dinosaur evolution, but sauropodomorphs were still absent. Some typical Late Triassic vertebrates were also a part of this biota, including sarcopterygians, temnospondyls, phytosaurs (known from isolated teeth), a large archosaur predator (a rauisuchian known from teeth and isolated bones), abundant turtles (*Proterochersis*) and aetosaurs (*Kocurypelta*). In addition, characteristic pentadactyl footprints (*Pentasauropus*) and oval-shaped, plant-rich bromalites from Poręba suggest the presence of medium-sized to large dicynodonts. Terrestrial turtles appear in the Poręba–Kocury assemblage, but have not been found in other fossil communities in the area.

In contrast to the Carnian–Norian biotas, no phytosaur, aetosaur, rauisuchian or silesaurid remains are known from the fossil record of the slightly younger, late Norian–earliest Rhaetian, sites at Lisowice and Marciszów (Fig. 1 and Supplementary Information). Instead, the tetrapod components of the Lisowice–Marciszów biota comprised the giant dicynodont *Lisowicia*, the large theropod-like *Smok*, small and medium-sized theropods, an omnivorous/herbivorous early dinosaur (supposedly an early ornithischian), a variety of small diapsids/archosauromorphs (for example, a sphenodont, a thalattosaurid, a crocodylomorph, gliding/flying reptiles), a eucynodont, a mammaliaform (*Hallautherium*) and temnospondyls (*Gerrothorax, Cyclotosaurus*). The ichnological record of this biota is mainly represented by tracks of large dicynodonts (*Pentasauropus*), small to large theropods (*Grallator, Anchisauripus, Kayentapus* and *Eubrontes*) and early ornithischians (*Anomoepus*)[32]. There is thus an overlap between the bone and trace records in the Lisowice–Marciszów assemblage. No sauropodomorph fossils (bones or tracks) have so far been found in association with the Lisowice–Marciszów assemblage, perhaps suggesting that they had not colonized the Polish Basin by the late Norian–earliest Rhaetian (approximately 205–210 million years ago (Ma)). The absence of sauropodomorphs stands out, as they had inhabited most of the rest of the Central European Basin since the mid–late Norian[24,33].

The first definitive appearance of herbivorous sauropodomorphs in the Polish Basin is instead evidenced by small *Evazoum* and large *Tetrasauropus*-like tracks from the mid–late Rhaetian Gromadzice–Rzuchów assemblage (Fig. 1 and Supplementary Information). This biota is known only from a modest fossil record, but shows that sauropodomorphs lived alongside crocodylomorphs (*Batrachopus*-like tracks), early ornithischians (small *Anomoepus* tracks) and small to large theropod dinosaurs (*Grallator, Anchisauripus* and *Eubrontes* tracks). Rare skeletal fossils (cranial fragments and teeth) suggest that some temnospondyls were still present (*Cyclotosaurus*), but no other characteristic Late Triassic tetrapods, such as phytosaurs or pseudosuchians, are known from this mid–late Rhaetian site of the Polish Basin.

We reconstructed food webs using this direct evidence for trophic interactions (bromalites and bite marks), complemented with comparative anatomy and functional morphology data for taxa lacking bromalite records (Fig. 3 and Supplementary Information). The composition of the vertebrate assemblages varies between the sites, but the inferred diets and trophic interactions have many similarities. Terrestrial and aquatic food webs were interconnected in the Late Triassic. Small to medium-sized carnivores based their diets mostly on fish and, at least in Krasiejów–Woźniki, also on insects[34,35] (Fig. 2 and Extended Data Figs. 1–7 and 10). In bromalites from Poręba and Lisowice, the presence of biomarkers such as phytanic and pristanic acids, which are characteristic constituents of fish oil, further highlights how common piscivory was in these assemblages. Positive evidence of tetrapod prey is relatively rare and attributable to, at most, a single terrestrial top predator in each assemblage: the rauisuchian *Polonosuchus* of the Krasiejów–Woźniki biota and the osteophagous theropod-like archosaur *Smok* of the Lisowice–Marciszów biota[36] (Fig. 2). Some bromalites from Krasiejów and Woźniki contain plant remains (Supplementary Information), but plant-rich herbivore bromalites are more commonly found in the post-Carnian sites Poręba, Lisowice, Marciszów and Gromadzice–Rzuchów.

Large, elongated, non-spiral bromalites from Krasiejów that are full of fish remains (Extended Data Fig. 3) were probably produced by *Paleorhinus*. This phytosaur possessed an elongated rostrum with long, sharp teeth adapted for piscivory. Other fish-bearing bromalites are assigned to hybodont sharks, actinopterygians, lungfishes and temnospondyls, on the basis of the content, internal structure, size and morphology (for example, presence/absence of spirals) of the bromalites. Some elongated bromalites from Krasiejów are attributed to *Silesaurus* and indicate that this animal mostly ingested insects, especially beetles[34,35]. However, other remains of fish and plants show that these constituted at least occasional, or possibly seasonal, meals (Supplementary Information). The aetosaur *Stagonolepis* shows some adaptations for a scratch-digging feeding ecology[37], and its dentition suggests that it was herbivorous or omnivorous. Elongated plant-bearing bromalites, which contain numerous plant cuticles, wood remains and palynomorphs, that can putatively be assigned to aetosaurs were collected in Krasiejów, Woźniki and Poręba (Supplementary Information). A few specimens of oval-shaped bromalites, rich in organic particles and small plant remains, have also been found in both Krasiejów and Woźniki. They seem to represent droppings of a relatively large herbivore, most likely a dicynodont (Supplementary Information). Two large bromalites from Krasiejów contain partly dissolved bone remains of tetrapods. These were most likely produced by *Polonosuchus*, a large rauisuchian and top predator of the mid–late Carnian, which is also associated with bite marks on aetosaur bones. However, most bromalites associated with the Krasiejów–Woźniki biota derive from secondary and tertiary consumers that fed on fish and/or insects. Some are spiral and were thus produced by animals with a spiral gut valve (hybodont sharks, actiopterygians and sarcopterygians). These contain fish bones and scales, bivalves and arthropods[34,35,38]. Non-spiral bromalites, which have not been possible to assign to specific producers, bear evidence that several aquatic and terrestrial tetrapods of different sizes fed on fish, insects and/or plants (Supplementary Information).

Abundant bromalites attributed to *Lisowicia*[39], the youngest and largest known dicynodont in the fossil record[32,40], suggest that it had a restricted diet, feeding principally on conifers (Supplementary Information). Numerous cuticle fragments (mainly resistant *Brachyphyllum* remains) were also extracted from bone-bearing bromalites of a large theropod-like predator (*Smok*), and were probably involuntarily ingested while feeding. Moreover, plant remains in elongated bromalite specimens from Lisowice indicate the presence of another much smaller terrestrial herbivore, probably the early ornithischian dinosaur known from the skeletal and track record of the site. Several small to medium-sized theropods are known from the body and track fossil records of the Lisowice–Marciszów biota[32]; these are the most likely candidates

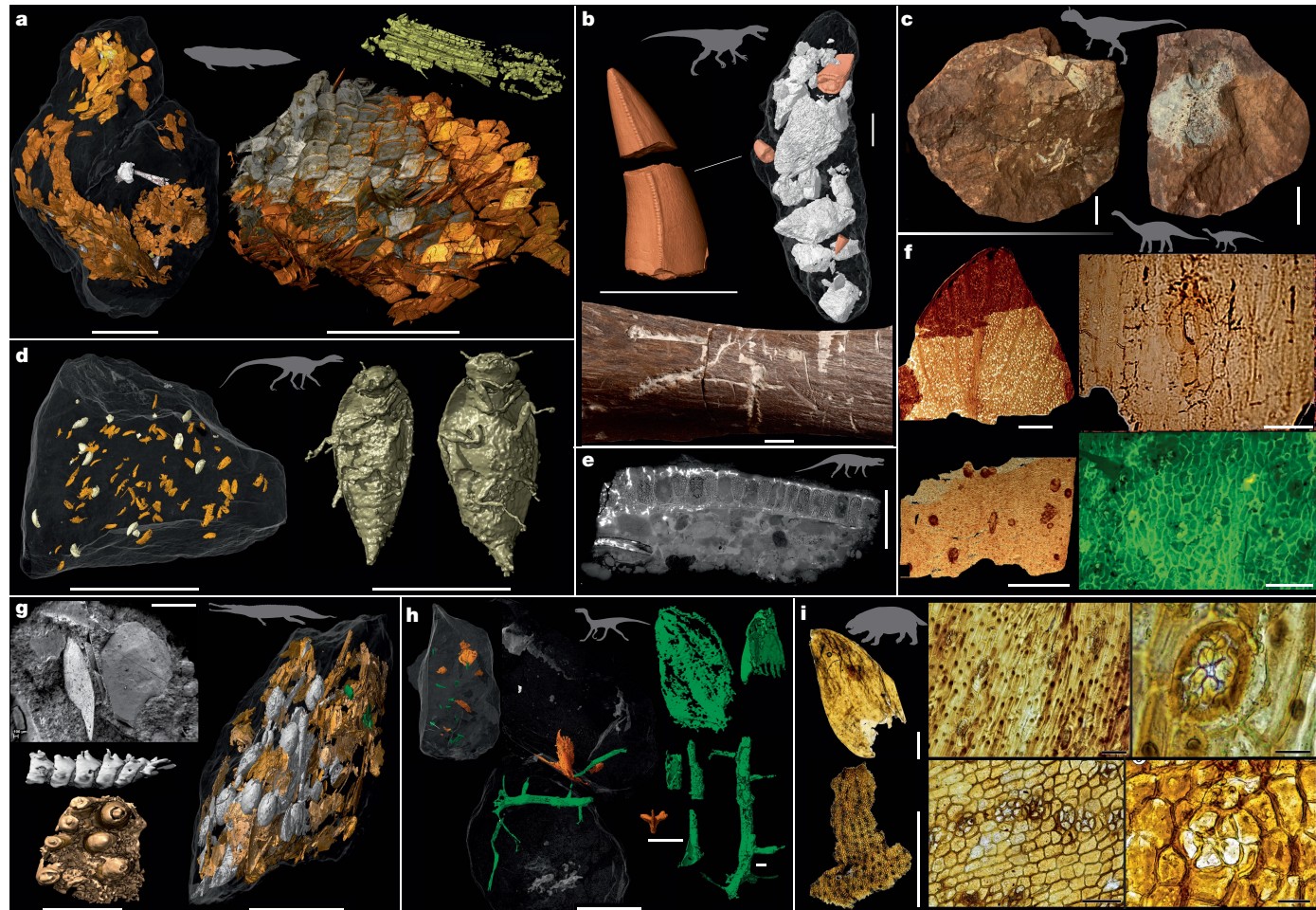

**Fig. 2 | Feeding evidence inferred from bite marks and synchrotron-scanned bromalites from Krasiejów, Lisowice and Sołtyków. a**, Big spiral bromalite (Institute of Paleobiology, Polish Academy of Sciences (ZPAL) AbIII/3401), presumably from the large dipnoan *Ptychoceratodus*[38], containing a semi-articulated fish (note the close-ups of articulated ganoid scales and a pelvic fin). **b**, Bite marks and bone-rich bromalites attributed to the archosaur *Smok*[36], exemplified by a dicynodont fibula with tooth marks (ZPAL V.33/471) and a coprolite containing teeth and bones (ZPAL V.33/471). **c**, Siderite bromalites (Polish Geological Institute−National Research Institute (MUZ PGI OS-221/300 and 306)), most likely produced by large predatory theropods, containing large bones including crocodylomorph limb bones. **d**, One of several insect-bearing *Silesaurus* bromalites (ZPAL AbIII/3520) with near-complete specimens of the beetle *Triamyxa coprolithica*[34,35]. **e**, Tooth-bearing temnospondyl bone from regurgitalite ZPAL AbIII/3417a, which also contains fish and supposed archosaur remains (Supplementary Information). Producer: *Polonosuchus*. **f**, Plant fossils from herbivore bromalites. Top left, *Komlopteris* pinna (Palaeozoic and Mesozoic of the National Biodiversity Collection−Herbarium KRAM at the W. Szafer Institute of Botany, Polish Academy of Sciences (KRAM) P PM 68/HS4/13). Top right, abaxial surface and stoma of *Desmiophyllum* KRAM P PM 68/HS5/25. Bottom left, gingkophyte cuticle with resin bodies, KRAM P PM 68/HS2/5. Bottom right, details of *Nilssonia* abaxial cuticle from fluorescence microscopy of KRAM P PM 68/PS6/10. See also Barbacka et al.[42] and the Supplementary Information. **g**, Fragment of big fish-bearing coprolite ZPAL AbIII/3440, probably produced by the phytosaur *Paleorhinus*, and close-ups of a tooth plate and articulated fish vertebrae. **h**, Fragments of plant-bearing elongated bromalites (ZPAL V.33/1203, ZPAL V.33/1206 and ZPAL V.33/1037). **i**, Plant fragments from acid-dissolved dicynodont bromalites (ZPAL V.33/1107-1109). The bromalites derive from Krasiejów (**a**, **d**, **e**, **g**), Lisowice (**b**, **h**, **i**), and Sołtyków (**c**, **f**). Scale bars, 10 mm (**a**, **b**, **c**, **d** (left), **e**, **g** (right)); 1 mm (**d** (right), **f** (left), **g** (inset), **g** (left), **h** (all)); 0.5 mm (**i** (left)); 0.1 mm (**i** (middle)); 50 μm (**f** (right), **i** (right)). Credits: Images adapted with permission from: **a**, ref. 38, Springer Nature Limited; **b**, ref. 36, Springer Nature Limited; **d**, ref. 35, Elsevier.

responsible for producing the elongated bromalites containing fish and bone fragments. Bromalites with spiral structures also contain various fish remains, including lepidotrichia, teeth, bones, abundant scales and soft tissue remains. Some of the fish remains are semi-articulated in a manner similar to that previously described for a bromalite from the Carnian of Krasiejów[38] (Fig. 2a). Small spiral bromalites were most likely produced by hybodont sharks or actinopterygians and larger ones by dipnoans and coelacanths, all known from the body fossil record at Lisowice, which seem to have been exclusively piscivorous (with the exception of a partly durophagous lungfish[32]). Small non-spiral bromalites were most likely produced by small diapsids/archosauromorphs, eucynodonts or mammaliaforms, all known from skeletal fossils of the site.

Ichnofossils associated with the mid−late Rhaetian Gromadzice−Rzuchów section suggest that a slightly different fauna had appeared in the region just before the dawn of the Jurassic. Plant-rich bromalites and up to 40-cm-long sauropodomorph tracks indicate the presence of relatively large herbivorous dinosaurs. In addition, two fish-bearing bromalites, produced by medium-sized predators, were found in the same interval. These were perhaps produced by theropods, known from a high diversity of 10- to 30-cm-long tridactyl tracks (*Grallator*, *Anchisauripus*, *Kayentapus* and *Eubrontes*). The fauna also included early ornithischian dinosaurs (*Anomoepus* tracks), crocodylomorphs (*Batrachopus*-like tracks) and temnospondyls (*Cyclotosaurus* bones and teeth).

## Transitional Triassic−Jurassic food webs

The latest Rhaetian-early Hettangian localities at Sołtyków and Hucisko (Fig. 1 and Supplementary Information) provide evidence of a rich

dinosaur assemblage in the Polish Basin. The fauna is inferred from rare bone findings and a well-studied ichnological record. It consisted of theropods of various sizes (cf. *Stenonyx*, *Grallator*, *Anchisauripus*, *Kayentapus* and *Eubrontes*; cf. *Megalosauripus* tracks), at least two small ornithischians (*Anomoepus* and *Delatorrichnus* tracks), three medium-sized to large sauropodomorphs (cf. *Tetrasauropus*; cf. *Otozoum* and *Parabrontopodus* tracks), a large turtle (unnamed tracks), a small pterosaur (*Pteraichnus*-like tracks), a small crocodylomorph (*Batrachopus*), a small lepidosauromorph (*Rhynchosauroides* tracks), a medium eucynodont (*Dicynodontipus* tracks) and a small mammaliaform (cf. *Ameginichnus* tracks)[7].

The bromalites from Sołtyków are very diverse in shape, phosphatic or secondarily mineralized (by siderite or pyrite), and measure from a few millimetres to more than 30 cm in length. Theropods, known from up to 55-cm-long tracks[41], probably produced the large bone-bearing bromalites. One of these contains skull and limb elements of an early crocodylomorph, which probably belong to the animal that produced the *Batrachopus* tracks (Fig. 2 and Supplementary Information). However, the menu of these large theropods probably extended far beyond crocodilians, as evidenced by the presence of fish scales and bone fragments of much larger prey items, which probably represent large sauropodomorph rib or limb fragments (Supplementary Information). Surprisingly, the large theropod bromalites also contain plant cuticles and palynomorphs of a high diversity, including plants previously unknown in the area. This possibly suggests that the large theropod, or its prey, had a big habitat or migrated across substantial distances[42]. Small to medium-sized elongated bromalites tentatively assigned to crocodylomorphs and theropod dinosaurs contain fish and small tetrapod remains. Moreover, many bromalites of various morphologies and geochemical compositions from a fine-grained lake interval suggest that fishes and small tetrapods were more abundant and diverse than the fossil record suggests.

Elongated, oval-shaped or irregular bromalites produced by herbivores (sauropodomorphs and ornithischians) contain numerous well-preserved plant remains, which also are more diverse than the flora inferred from the host rocks[42]. The affinities of the cuticles suggest that the herbivorous dinosaurs fed on seed ferns, cycads, ginkgophytes and bennettitaleans, which grew in a floodplain environment[42]. This is drastically different from the one-sided, conifer-dominated content of the Late Triassic herbivore bromalites. Charred wood, probably originating from widespread wildfires[43–45], was also commonly ingested by the earliest Jurassic herbivores, but not carnivores, as suggested from pyrolytic geochemical signatures and charcoal (inertinite) bromalite inclusions (Supplementary Information and Supplementary Tables 10–12). The burnt wood was probably ingested accidentally with unburnt parts of the plants after wildfires, or intentionally ingested for detoxification[46,47]. The *n*-alkanes from the bromalite samples suggest a feeding environment located in a rainforest climate regime, and common biomarkers include those from various conifer families, bacteria and ectomycorrhizal or wood-rot fungi[48–50] (Supplementary Information section 3.9).

## Discussion

The Late Triassic to earliest Jurassic interval was characterized by global climatic changes and an episode of extensive volcanism in the Central Atlantic magmatic province (CAMP)[51–53]. Large-scale tectonic processes on Pangaea, and the resulting shift of the position of the Polish Basin northwards in the latest Triassic, was the main factor behind the environmental changes in the area. The northward drift may have contributed to the termination of regional aridity, as has been suspected for contemporary successions in Greenland[54] and Sweden[55]. Data from the Polish Basin reflect a persistent subtropical warm and dry climate during most of the Late Triassic, with humid phases occurring in the middle Carnian (the Carnian Pluvial Event), the late Norian and the late Rhaetian[5–7] (Fig. 1 and Supplementary Information). On a broader scale, the Late Triassic environments in central Europe were subjected to a marked change from seasonally arid continental (Carnian–middle Norian) to permanent humid (mid–late Rhaetian–early Hettangian) conditions, coincident with the opening of internal seaways into the Pangaean interior and the marine inundation of central Europe. The increased humidity allowed a diversified vegetation cover to develop. These large-scale climatic trends led to environmental changes and reconfigurations of the floral assemblages in the region[7,25–28,56], which in turn had large effects on the tetrapod communities (Fig. 1). Several rapid climatic events (hot and humid periods, separated by cooler and drier periods) can be distinguished in the Triassic–Jurassic boundary interval in the Polish Basin[7]. The late Rhaetian interval of these events coincides with negative $\delta^{13}C_{org}$ excursions, perturbation of the osmium isotope system (attributed to volcanic fallout), polycyclic aromatic hydrocarbon occurrences and a turnover in the palynoflora, which suggest influences of the CAMP flood basalts on the Polish Basin and its ecosystems. These abrupt, short events are recorded on a regional scale in the basin, but it is more difficult to determine their direct impact on local faunas. The transition from Krasiejów–Woźniki to Sołtyków–Hucisko plant assemblages coincided with the disappearance of many typical Triassic groups of non-dinosaurian tetrapods. This is captured in the skeletal and footprint fossil record, but also in the diversity and content of bromalites. It is noticeable that the bromalite fossil record reflects the faunal turnovers in the late Norian–earliest Rhaetian and middle Rhaetian–early Hettangian intervals (Fig. 1 and Supplementary Information). Food residues extracted from dicynodont and aetosaur bromalites support feeding habits considerably different from those of the newly emerging herbivorous dinosaurs. As specialists, aetosaurs and dicynodonts were potentially more constrained than the dinosaur herbivores in shifting their diets towards the new prevailing flora. As a result, the terrestrial herbivore guild was completely replaced by sauropodomorphs and ornithischians by the mid–late Rhaetian–earliest Hettangian. There are many indications that herbivorous sauropodomorphs did not adapt to new conditions locally, but rather migrated, in the mid–late Rhaetian time, to the Polish Basin as soon as ecological conditions allowed it. The first sauropodomorphs appear in the fossil record of the Northern Hemisphere temperate belt about 214 Ma[23]. This dispersal was related to a concomitant attenuation of climate barriers, but it can also be speculated that this migration may be the result of the search for new suitable habitats. The timing and abruptness of these Late Triassic faunal changes throughout Pangaea have been much discussed, and it has become increasingly clear that the diversification and dispersal of early dinosaurs were complex and influenced by climatic factors[18–24,57]. The fossil record from the Polish Basin shows that it was during the late Norian–earliest Rhaetian interval (and not during end-Carnian, mid-Norian or end-Triassic events) that phytosaurs, rauisuchians, aetosaurs, dicynodonts and other Triassic groups experienced the greatest reduction in diversity. The disappearance of these formerly dominant tetrapods is mirrored by an increased abundance of dinosaurs in the body and trace fossil records[58] (Fig. 1 and Supplementary Information). Based on our model in a restricted geographical setting, we show that the rise of dinosaurs to ecological dominance can be broken down into five phases (Fig. 3), marking the appearance of:

1. small, opportunistic and omnivorous dinosaur precursors in the mid–late Carnian;
2. fauna with dinosaur precursors and the first predatory dinosaurs in the mid–late Norian;
3. diversified small to medium-sized predatory theropods, as well as the first large theropods and omnivorous/herbivorous ornithischians in the late Norian–earliest Rhaetian;
4. small ornithischians and medium-sized to large herbivorous sauropodomorphs and diversified theropods in the mid–late Rhaetian and finally;

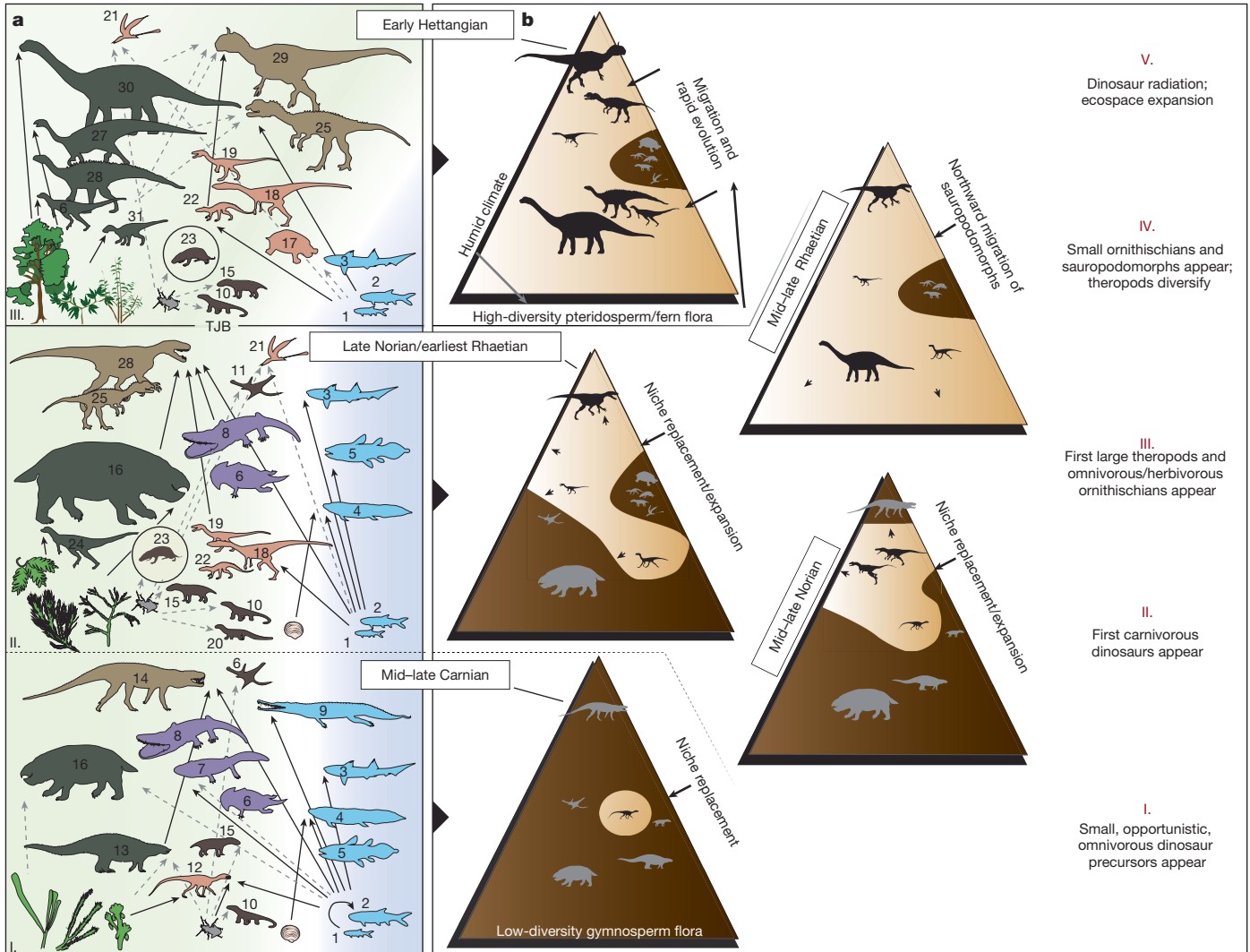

**Fig. 3 | Reconstructed food webs across the Triassic–Jurassic transition, model for the dinosaurs' stepwise rise to dominance and key phases of dinosaur evolution in the Polish Basin. a**, Food webs from the mid–late Carnian Krasiejów–Woźniki biota (bottom), the late Norian–earliest Rhaetian Lisowice–Marciszów biota (middle) and the earliest Hettangian Sołtyków–Hucisko biota (top). Black arrows indicate trophic relationships inferred from direct evidence of feeding (bromalites and bite marks). Grey dotted lines indicate trophic relationships inferred from indirect ecological evidence such as comparative anatomy and functional morphology. Vertebrate guilds: light brown, terrestrial top predators; orange, small to medium-sized terrestrial carnivores; brown, small insectivores; green, terrestrial herbivores; purple,

near-shore piscivores; and light blue, pelagic carnivores. See Fig. 1 and Supplementary Table 2 for taxon identities. Floras (Supplementary Information) are exemplified by the following: I, *Sphenopteris*, *Glyptolepis* and *Pterophyllum* (*Voltzia* floral assemblage); II, *Brachyphyllum*, *Podozamites* and *Lepidopteris* (*Brachyphyllum* floral assemblage); and III, *Komlopteris*, *Nilssonia* and *Podozamites* (*Thaumatopteris* floral assemblage). **b**, Model for the stepwise rise of dinosaurs in the Polish Basin (north-central Pangaea) based on the trophic interactions in **a** and the two intermediate biotas Poręba–Kocury and Gromadzice–Rzuchów (the right column), and other data from the Polish Basin (main text, Extended Data Figs. 1–10 and Supplementary Information).

5. a dinosaurian diversification and expansion of ecospace occupation in the latest Rhaetian–earliest Hettangian.

The last two phases were probably the result of two superimposed processes: environmental changes and the appearance of new dietary key adaptations that enabled exploitation of the new food resources in an unprecedented way (Fig. 3). The biggest change in trophic dynamics, specified by a large diversity increase in bromalite shape and contents, occurred in the latest Rhaetian–earliest Hettangian, the time interval marked globally by massive volcanism, the end-Triassic extinction and the immediate early recovery following it. It is difficult to assess what direct impact these events had on the evolution of dinosaur diversity and trophic complexity, but the timing of the events suggests that there was a complex interplay of several processes: a degree of opportunism coupled with anatomical differences or increased phenotypic

plasticity that enabled herbivorous dinosaurs to better cope with the environmental changes.

Despite the biases and uncertainties of the fossil record (for example, selective preservation/sampling of rocks, animals, tissues and environments), we demonstrate that integrated analyses of body fossils, tracks and bromalites (Supplementary Tables 2–9 and 13) provide robust pictures of past food webs, casting new light on early dinosaur evolution and the origin of the first complex dinosaur faunas with megaherbivores and predators. The apparent changes in bromalite morphologies and their contents are easily explained given the shift in faunal composition and tetrapod diversity across the Late Triassic and earliest Jurassic interval. These conclusions are further supported by the fact that the studied bromalites originate from similar sedimentary environments, and that herbivore bromalites, which are typically rare, have been recovered from all biotas (Supplementary Table 13 and

Extended Data Figs. 1–4, 7 and 9). Our data suggest that climatic and environmental changes and the ensuing substantial transformations in vegetation were an important stimulus for the development of dinosaur faunas in the Late Triassic ecosystems of the Polish Basin. The data also indicate that the main environmental changes that occurred at the very end of the Triassic period paved the way for the early radiation and increased abundance of dinosaurs. Our results support the idea that stochastic processes coupled with a competitive advantage enabled the enormous evolutionary success of dinosaurs. In sum, the dinosaurs rose to supremacy in a stepwise fashion across 30 million years of evolution. Thanks to an increased resolution of the fossil record and stratigraphical control in regional basins across this time, we can use data to describe these steps, here presented as five distinct phases, which we believe can also describe global patterns (Fig. 3b).

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

## Methods

### Fieldwork and data collection

The bromalites derive from natural or artificial Upper Triassic and Lower Jurassic site exposures located in Silesia and the Holy Cross Mountains in the Polish Basin area (Supplementary Information and Supplementary Figs. 1–9 in ref. 60). A total of 532 bromalites have been collected from eight fossiliferous sites (Supplementary Tables 2–9). The specimens were collected between 1996 and 2017 by G.N., T.S., K.O., G. Pieńkowski and M.Q. All bromalites are stored in the scientific collections of the Polish Geological Institute–National Research Institute (Warszawa, Kielce; acronym Muz. PGI; Muz. PGI OS), Institute of Paleobiology, Polish Academy of Sciences (Warszawa; acronym ZPAL). Specimens were photographed and measured; some of them, representing characteristic morphotypes, are presented in illustrations (Supplementary Figs. 14–26 in ref. 60).

### Optical microstructure observations

Numerous bromalites were studied in detail in thin sections. Standard petrographic thin sections were prepared and examined under an optical microscope (Nikon Eclipse LV100 POL and Leica DM). Images were collected using digital cameras.

### Phase-contrast synchrotron microtomography

More than 100 bromalite specimens from Krasiejów and Lisowice (Supplementary Tables 3 and 6) were scanned using propagation phase-contrast synchrotron microtomography as part of project ES145 at beamline ID19 of the European Synchrotron Radiation Facility in Grenoble, France. Specimens were selected for synchrotron scans after initial inspection showing that they contained well-preserved inclusions. Specimens from Woźniki, Poręba, Gromadzice–Rzuchów, Hucisko and Sołtyków were not synchrotron scanned (three specimens from Sołtyków were subjected to computed tomography, but the results are not presented here).

Different scan settings were applied to the bromalites (depending on their size) to maximize resolution, while still maintaining a field of view that would capture an entire specimen; the specific parameters for all the four different scan settings that were used are provided in Supplementary Table 1. For all scans, the propagation distance (that is, the distance between the sample on the rotation stage and the camera) was 2,800 mm and the camera was a sCMOS pco.edge 5.5 detector. Reconstructions of the scanned data were based on a phase-retrieval approach[61–63]. Ring artefacts were corrected by using an in-house correction tool[63]. Binned versions (bin factor of 2) were calculated for fast processing and screening of the samples. The final volumes consist of stacks of JPEG2000 or TIFF images that were imported and segmented in the software VGStudio MAX v.3.1 (Volume Graphics Inc.).

### Identification of the excrement producer

The bromalites were classified according to their gross morphology, but also their composition and inclusions (Fig. 1 and Supplementary Tables 3–9). On the basis of the comparison with other Late Triassic and Early Jurassic bromalites, and considering features such as shape, size, content and stratigraphic provenance, we identify which taxon/ichnotaxon was the most likely producer. These data were complemented with comparative anatomy, general feeding ecology and functional morphology data for taxa/ichnotaxa recorded (skeletal data and identified trackmakers) from the assemblages (Supplementary Information and Supplementary Fig. 10 in ref. 60).

### Analysis of plant cuticles

Bromalites were treated with 10% hydrochloric acid (HCl) and 20% hydrofluoric acid to remove sediment. Retrieved cuticles were macerated using Schultze's reagent (25% nitric acid with a few crystals of potassium chlorate) and subsequently treated with 5% potassium hydroxide. The residue was rinsed with water after each step. Macerated cuticles were washed with distilled water and dehydrated in pure glycerine. Each cuticle was kept in an Eppendorf tube in pure glycerine with a few drops of thymol to prevent fungal growth. Cuticles were analysed with a Nikon Eclipse 50i transmitted light microscope and documented with a Nikon DSFi2 digital camera and Nikon NIS-Elements imaging software. Selected cuticles were analysed using a Philips XL30 environmental scanning electron microscope (SEM).

Measurements of length, width and surface area of the cuticles were made using the NIS-Elements software. The length of a cuticle was measured according to the arrangement of epidermal cells that indicate a top and a base of an organ (for example, leaf or seed) between the most distal ends. The width of a cuticle was measured perpendicular to the length. Because of the irregular shapes of the cuticles, the surface area was estimated using the NIS-Element tool 'ellipse' for measuring surface.

### Scanning electron microscopy coupled with energy-dispersive X-ray spectroscopy

Bromalite fragments from Krasiejów (5 specimens), Poręba (3 specimens), Lisowice (6 specimens) and Sołtyków (11 specimens) were broken off, glued onto stubs and coated with platinum or gold. Material was analysed in a Philips XL20 SEM equipped with the energy-dispersive detector ECON 6, system EDX-DX4i and a backscatter electron detector for Compo or Topo detection (FEI product). This instrument was operated at an accelerated voltage of 25 kV, a beam current of 98–103 nA and a spot diameter of 4 μm. SEM images were collected.

### Total organic carbon and total sulfur measurement

The total carbon, total sulfur and total inorganic carbon (TIC) contents were determined using an Eltra CS-500 IR analyser with a TIC module (at the Faculty of Natural Sciences, the University of Silesia, Katowice, Poland). The total organic carbon content was calculated as the difference between total carbon and TIC. An infrared cell detector of $CO_2$ gas was used to measure the content of total carbon and TIC, which was evolved by combustion under an oxygen atmosphere for total carbon, and was obtained from reaction with 10% HCl for TIC. The standards used for the calibration were from Eltra. Analytical precision and accuracy were as follows: plus or minus 2% for total carbon and plus or minus 3% for TIC.

### Organic petrology

Seven samples were selected for petrological observations (SOL1 a, b and c, SOL_3, SOL_4, SOL_5 and SOL_7). Vitrinite and fusinite reflectance for macerals from Sołtyków have been published elsewhere[43]. The sample preparation process follows the procedure described in ISO 7404-2 (2009). Microscopic examination of the samples in reflected light and immersion oil was performed using an optical microscope Axio Imager.A2m (Faculty of Natural Sciences, the University of Silesia, Katowice, Poland).

### Extraction, separation and derivatization

The extraction with DCM (dichloromethane)/methanol (MeOH) (50:50, v:v) was done with a Dionex 350 Accelerated Solvent Extractor system (Thermo Scientific) at 80 °C in 34 ml stainless steel cells (pressure ($p$) = 10 MPa, solvent flow = 70 ml min$^{-1}$). Each extract was concentrated and separated into three fractions, aliphatic, aromatic and polar, using micro-column chromatography[64]. The silica gel used for separation of a particular fraction had been activated at 120 °C for 24 h. The following elution method was applied: (1) $n$-pentane (aliphatic fr.), (2) $n$-pentane and DCM (7:3, v:v – aromatic fr.) and (3) DCM/MeOH (1:1, v:v – polar fr.). The polar fraction of seven selected samples was derivatized to trimethylsilyl derivatives by reaction with $N,O$-bis-(trimethylsilyl)trifluoroacetamide (BSTFA), 1% trimethylchlorosilane (Sigma-Aldrich) and pyridine (Sigma-Aldrich) for 3 h at 70 °C. Fractions were analysed by gas chromatography–mass spectrometry (GC–MS). Internal standards

(ethyl vanillin, phenylindene) were added to the total extracts. A blank sample (silica gel) was analysed using the same procedure (including extraction and separation on columns). Only trace amounts of fatty acids and phthalates were found in the blank.

## Gas chromatography–mass spectrometry

GC–MS analyses were carried out using an Agilent Technologies 7890A gas chromatograph and an Agilent 5975C Network mass spectrometer with triple-axis mass selective detector (MSD). Helium (6.0 grade) was used as a carrier gas at a constant flow of 2.6 ml min$^{-1}$. The separation was obtained on a fused silica capillary column (J&W HP5-MS, 60 m × 0.25 mm i.d., 0.25 μm film thickness) coated with a chemically bonded phase (5% phenyl, 95% methylsiloxane), for which the gas chromatography oven temperature was programmed from 45 °C (1 min) to 100 °C at 20 °C min$^{-1}$, then to 300 °C at 3 °C min$^{-1}$ (hold 60 min), with a solvent delay of 10 min. The gas chromatography column outlet was connected directly to the ion source of the MSD. The GC–MS interface was set at 280 °C; the ion source and the quadrupole analyser were set at 230 °C and 150 °C, respectively. Mass spectra were recorded from $m/z$ 45 to 550 (0–40 min) and $m/z$ 50–700. The mass spectrometer was operated in the electron impact mode, with an ionization energy of 70 eV. All GC–MS analyses were performed at the Faculty of Earth Sciences, Sosnowiec. An Agilent Technologies MSD ChemStation E.02.01.1177, the Wiley Registry of Mass Spectral Data (tenth edition) and NIST 17 software were used for data collection and mass spectra processing.

## Samples

Geological and palaeobotanical samples (Supplementary Figs. 11–13 and 27–31 in ref. 60), all the studied bromalite specimens and bone with bite marks (Supplementary Fig. 32 in ref. 60) are housed in the scientific collection at the Polish Geological Institute—National Research Institute (Warszawa, Kielce); at the Institute of Paleobiology, Polish Academy of Sciences (Warszawa); in the collections of research results at the University of Silesia (Sosnowiec; palaeobotanical data); in the palaeobotanical collection Palaeozoic and Mesozoic of the National Biodiversity Collection—Herbarium KRAM at the W. Szafer Institute of Botany, Polish Academy of Sciences, Cracow, Poland; and at Jagiellonian University (Kraków; palaeobotanical data).

## Reporting summary

Further information on research design is available in the Nature Portfolio Reporting Summary linked to this article.

## Data availability

Reconstructed image stacks of the synchrotron-scanned bromalites are publicly available in ESRF's heritage database for palaeontology, evolutionary biology and archaeology: https://paleo.esrf.eu/explore/ichnology/Coprolites (https://doi.org/10.15151/ESRF-DC-1848198683, https://doi.org/10.15151/ESRF-DC-1848198699, https://doi.org/10.15151/ESRF-DC-1848198691, https://doi.org/10.15151/ESRF-DC-1848199407, https://doi.org/10.15151/ESRF-DC-1848199415, https://doi.org/10.15151/ESRF-DC-1848199431, https://doi.org/10.15151/ESRF-DC-1848199423, https://doi.org/10.15151/ESRF-DC-1823716285, https://doi.org/10.15151/ESRF-DC-1823716293, https://doi.org/10.15151/ESRF-DC-1823716301, https://doi.org/10.15151/ESRF-DC-1848198659, https://doi.org/10.15151/ESRF-DC-1848198675, https://doi.org/10.15151/ESRF-DC-1848198667). Supplementary figures are available at Figshare (https://doi.org/10.6084/m9.figshare.26103031)[60].

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

**Acknowledgements** The bromalites were scanned as a part of project ES145 at beamline ID19 of the European Synchrotron Radiation Facility (ESRF). Many thanks to P. Tafforeau (ESRF) for all the help during the scanning process and reconstructions of the data. Thanks to D. Snitting (Uppsala University) for technical support and scientific discussions. Additional thanks to S. Huld and M. A. D. During (Uppsala University), who helped us with virtual reconstructions of several bromalite specimens, and P. Bajdek (Częstochowa, Poland) who provided fruitful discussions on the bromalite external morphologies. The results are published with the permission of the Polish Geological Institute—National Research Institute (PGI-NRI) (Warsaw) and Institute of Paleobiology, Polish Academy of Sciences (PAS) (Warsaw). M.Q. was funded by a postdoctoral grant (grant no. 2020-06445) from the Swedish Research Council. G.N. was funded by a grant (grant no. 2017-05248) from the Swedish Research Council. This study was supported by the Polish Ministry of Science and Higher Education as project no. 3941/B/P01/2009/36 (2009–2014, study grant of G.N., Faculty of Biology, University of Warsaw); by funds from the National Science Centre, Poland (projects 2017/25/B/ST10/01273, 2012/06/M/ST10/00478 and 2014/15/N/ST10/05142); and by statutory funds of the W. Szafer Institute of Botany, Polish Academy of Sciences. It was also supported by a special grant from *National Geographic Polska* (2008–2009) and by a project grant from Uppsala University (2017). P.E.A. acknowledges the support of a Wallenberg Scholarship from the Knut and Alice Wallenberg Foundation.

**Author contributions** M.Q. and G.N. conceived the idea, designed the project, analysed the data and prepared the paper together. M.Q. and G.N. performed the scanning. M.Q. and J.V.W. segmented the three-dimensional data. M.Q. drafted, together with G.N., the first version of the manuscript, which P.E.A. modified. G.N., T.S. and G. Pieńkowski organized fieldwork and collected material. Data were also analysed by J.V.W. (bromalites), Z.W. (palaeobotany), M.B. (palaeobotany), G. Pacyna (palaeobotany), A.G. (palaeobotany), J.Z. (palynology), A.J. (palaeobotany), K.O. (taphonomy), T.S. (vertebrate palaeontology), L.M. (geochemistry) and G. Pieńkowski (regional geology).

**Funding** Open access funding provided by Uppsala University.

**Competing interests** The authors declare no competing interests.

**Additional information**

**Correspondence and requests for materials** should be addressed to Martin Qvarnström or Grzegorz Niedźwiedzki.

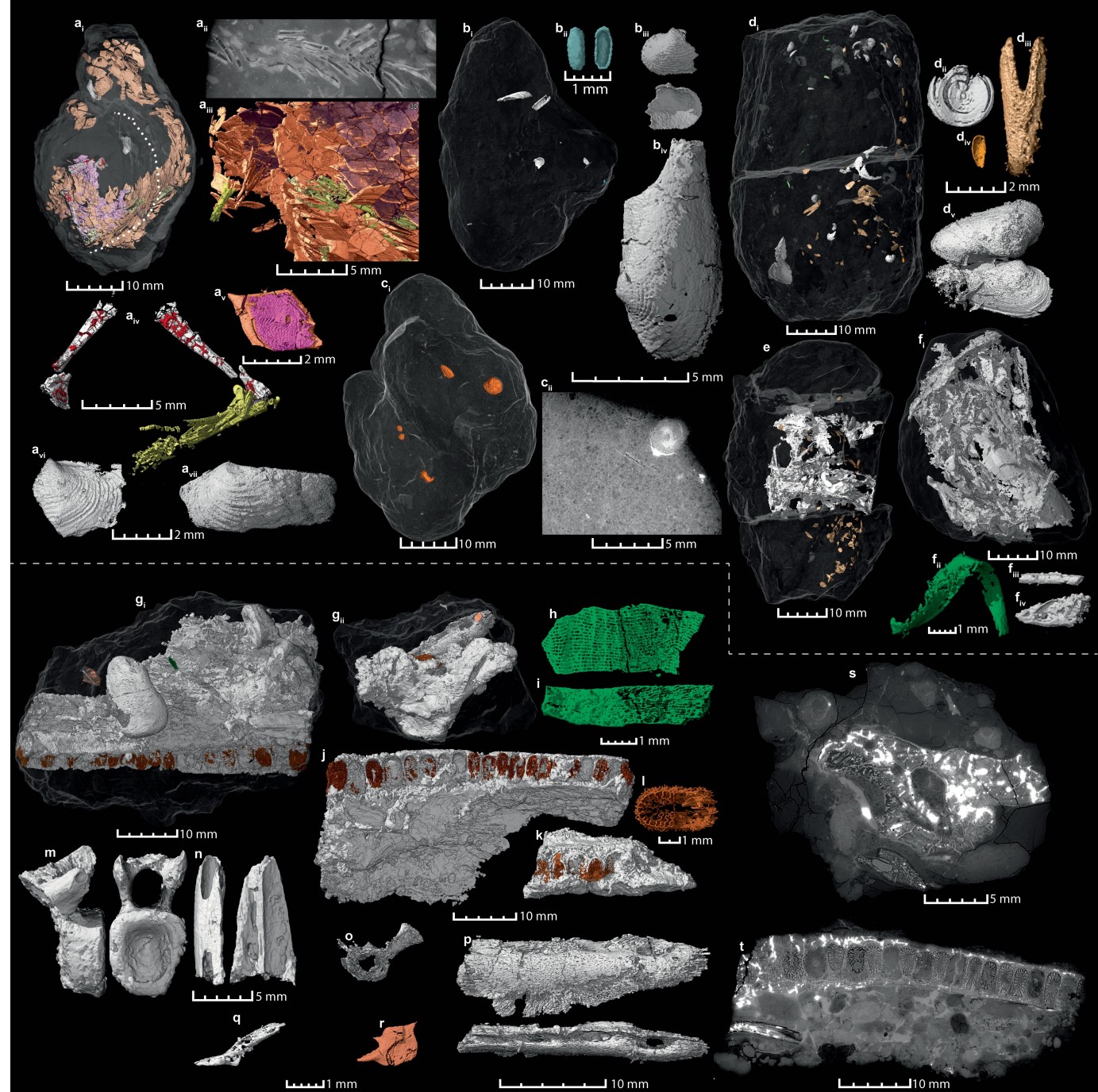

**Extended Data Fig. 1 | 3D-reconstructed contents from synchrotron-scanned scroll coprolites attributed to *Ptychoceratodus* (a-f) and probable regurgitalites attributed to *Polonosuchus* (g-t) from Krasiejów. $a_i$,** Specimen ZPAL AbIII_3401 in semi-transparency with internal fish remains ($a_{ii}$-$a_v$) and crushed bivalves ($a_{vi}$-$a_{vii}$). The dotted line marks a big spiral convolution. **$a_{ii}$,** Virtual thick slab of the coprolite matrix and articulated fish scales. **$a_{iii}$,** Close up of semi-articulated fish scales (the bony part of the scale is highlighted in orange and the ganoine layer in pink) and fin lepidotrichia (green). **$a_{iv}$,** Pelvic girdles and articulated right pelvic fin. **av,** Example of a fish scale. Note the lateral line canal opening (centre). **$a_{vi}$, $a_{vii}$,** Crushed bivalve shells. **$b_i$,** Specimen ZPAL AbIII/3413 in semi-transparency with internal crushed bivalves. **$b_{ii}$,** Ostracod valve in lateral views. **$b_{iii}$,** External and internal view of a crushed shell. **$b_{iv}$,** External view of a crushed bivalve shell. **$c_i$,** Specimen ZPAL AbIII/3412 in semi-transparency with internal unidentified globules. **$c_{ii}$,** Virtual thin section showing the coprolite matrix and one of the globules. **$d_i$,** Specimen ZPAL AbIII/3416 in semi-transparency with inclusions. **$d_{ii}$,** Round inclusion (here preserved as a hemisphere) with internal spiral structure. **$d_{iii}$,** A mid-line fish scale.

**$d_{iv}$,** Beetle elytron. **$d_v$,** Two bivalve shells in articulation. **e,** Specimen ZPAL AbIII/3415 in semi-transparency with internal fish remains and mineralized cracks. **$f_i$,** Specimen ZPAL AbIII/3414 in semi-transparency with internal cracks and inclusions. **$f_{ii}$,** Plant inclusion. **$f_{iii}$, $f_{iv}$,** Small bone inclusions. **g,** Two fragments (**$g_i$** specimen ZPAL AbIII/3417a, **$g_{ii}$** Specimen ZPAL AbIII/3417b) rendered in semi-transparent. **h-i,** Plant cuticles. **j,** Tooth-bearing fragment from a temnospondyl skull. Note the folded dentine ('labyrinthodont' plicidentine) which is the characteristic tooth structure of early tetrapods including temnospondyls. **k,** A smaller fragment of a tooth-bearing fragment of a temnospondyl skull (from specimen **$g_{ii}$**, but perhaps representing a fragment of the same bone as in **$g_i$**). **l,** Detailed view on folded dentine in one of the teeth. **m,** A vertebra, likely of a tetrapod. **n,** Triangular bone with large canals (perhaps from a temnospondyl). **o,** Small, poorly preserved vertebra. **p,** Elongated bone with large canals (likely from a temnospondyl). **q,** Small fish bone. **r,** Ganoid scale of an actinopterygian fish. **s,** Virtual thin section displaying the bromalite matrix and several bones (including the small temnospondyl skull bone shown in **k**). **t,** Virtual thin section displaying the bromalite matrix and the large temnospondyl skull bone in **j**.

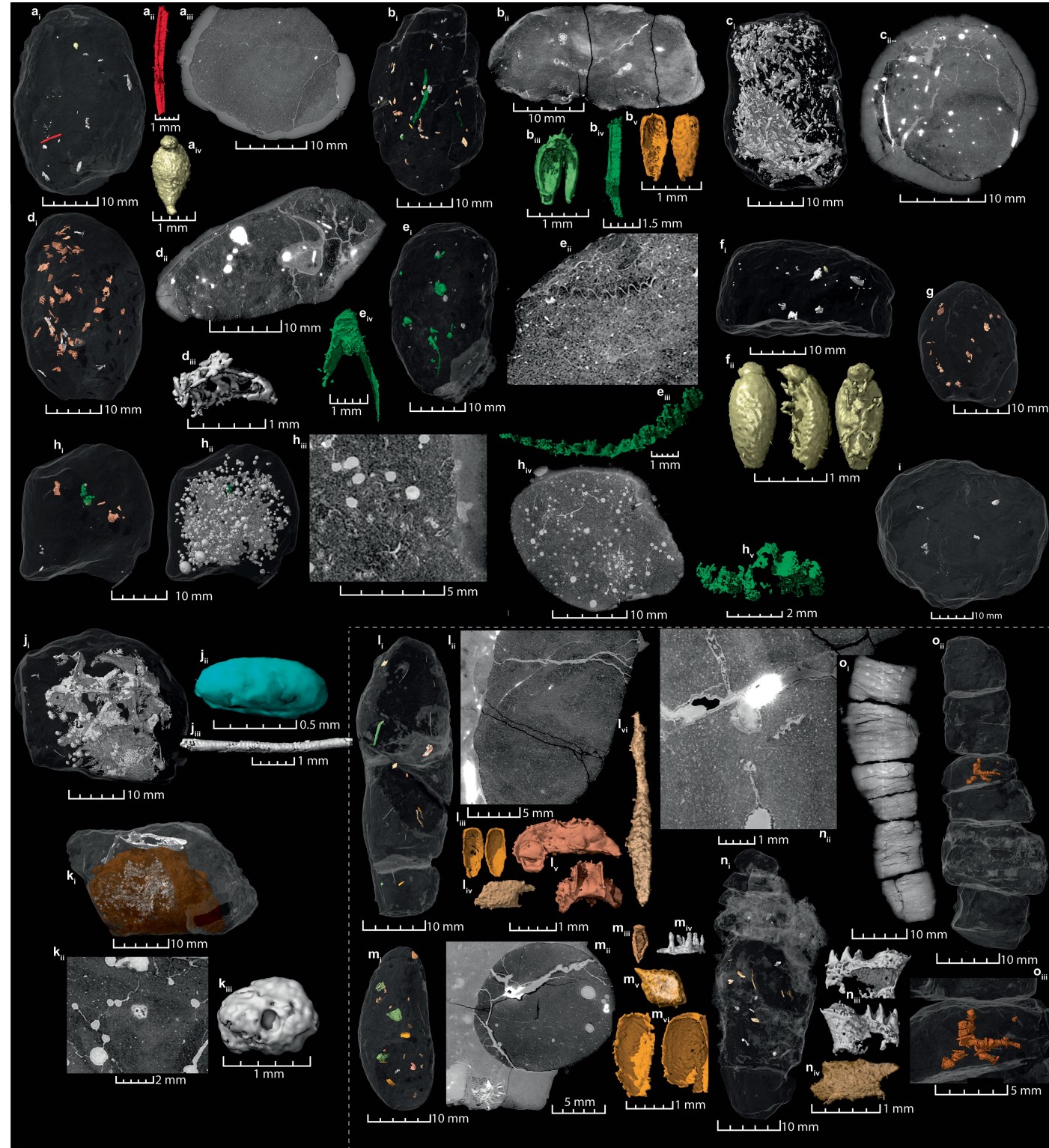

**Extended Data Fig. 2** | See next page for caption.

**Extended Data Fig. 2 | Virtual reconstructions of contents from oval (a-k) and elongated (l-o) coprolites from Krasiejów. $a_i$,** Specimen ZPAL AbIII/3439 in semi-transparency with inclusions. **$a_{ii}$,** Arthropod appendage. **$a_{iii}$,** Virtual thin section of the coprolite matrix **$a_{iv}$,** Beetle in ventral view. **$b_i$,** Specimen ZPAL AbIII/3419 in semi-transparency with visible inclusions. **$b_{ii}$,** Virtual thin section of the coprolite matrix. **$b_{iii}$,** Possible plant inclusion. **$b_{iv}$,** Plant inclusion. **$b_v$,** Beetle elytron. **$c_i$,** Specimen ZPAL AbIII/3420 in semi-transparency with numerous spicules of unknown origin. **$c_{ii}$,** Virtual thin section showing the dense spicules and coprolite matrix. **$d_i$,** Specimen ZPAL AbIII/3422 in semi-transparency with numerous fish scales. **$d_{ii}$,** Virtual thin section of the coprolite matrix. **$d_{iii}$,** Possibly a small bone. **$e_i$,** Specimen ZPAL AbIII/3426 in semi-transparency with numerous fish scales. **$e_{ii}$,** Virtual thin section of a wavy plant inclusion. **$e_{iii}$,** The wavy plant inclusion rendered in 3D. **$e_{iv}$,** Plant inclusion. **$f_i$,** Specimen ZPAL AbIII/3428 in semi-transparency with some dense mineralisations. The specimen is fragmentary and could represent a part of a bigger coprolite. **$f_{ii}$,** An articulated beetle specimen. **g,** Specimen ZPAL AbIII/3418 in semi-transparency with visible fish inclusions. **$h_i$,** Specimen ZPAL AbIII/3421 in semi-transparency with fish scales and plant inclusions and (**$h_{ii}$**) with spherical structures visible.

**$h_{iii}$,** Virtual thin section showing spherical structures and a wavy plant inclusion. **$h_{iv}$,** Virtual thin section. **i,** Specimen ZPAL AbIII/3424 in semi-transparency with a few unidentified inclusions. **$j_i$,** Specimen ZPAL AbIII/3427 in semi-transparency with numerous spherical structures, cracks and inclusions. **$j_{ii}$,** An ostracod in articulation. **$j_{iii}$,** Small bone fragment. **$k_i$,** Specimen ZPAL AbIII/3423 in concretion. **$k_{ii}$,** Virtual thin section showing mineralised bubbles and inclusion with a chamber and tube. **$k_{iii}$,** Inclusion with a chamber and tube. **$l_i$,** Specimen ZPAL AbIII/3429 in semi-transparency with fish and arthropod inclusions. **$l_{ii}$,** Virtual thin section showing the coprolite matrix. **$l_{iii}$,** Beetle elytra. **$l_{iv}$,** Fish scale. **$l_v$,** Bilateral structure of unknown origin. **$l_{vi}$,** Fish remain. **$m_i$,** Specimen ZPAL AbIII/3431 in semi-transparency with fish and arthropod inclusions. **$m_{ii}$,** Virtual thin section of the coprolite matrix with some spherical structures. **$m_{iii}$,** Fish or arthropod remain. **$m_{iv}$,** Possibly a pharyngeal ossicle from a branchial arch. **$m_v$,** Fish scale. **$m_{vi}$,** Beetle elytra. **$n_i$,** Specimen ZPAL AbIII/3432 in semi-transparency with fish scales. **$n_{ii}$,** Virtual thin section showing an inclusion with denticles. **$n_{iii}$,** Inclusion with denticles. **$n_{iv}$,** Fish scale. **$o_i$,** Virtual thin section and (**$o_{ii}$**) semi-transparent version of coprolite ZPAL AbIII/3430. **$o_{iii}$,** Some of the numerous invertebrate burrows in the coprolite.

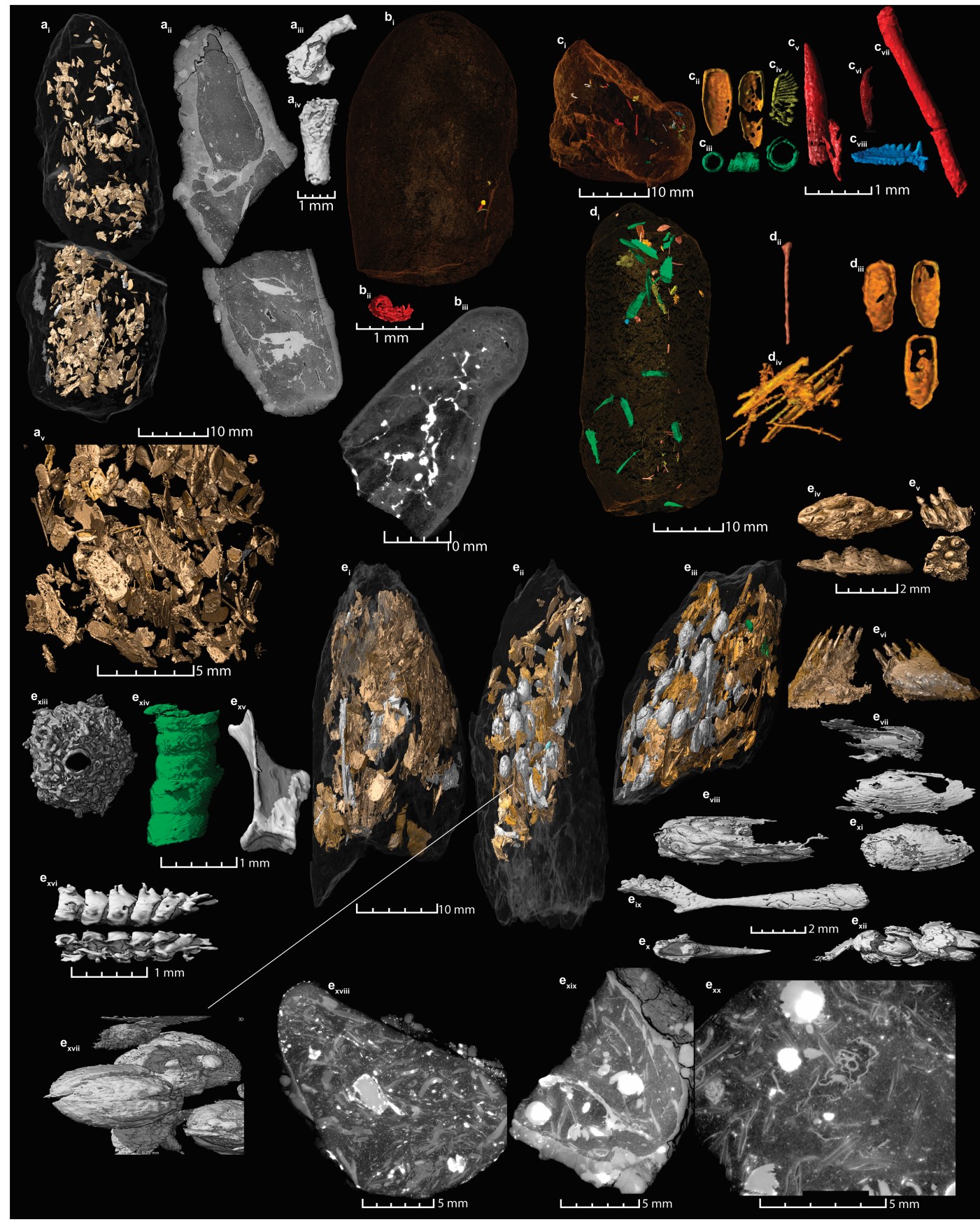

**Extended Data Fig. 3** | See next page for caption.

**Extended Data Fig. 3 | Virtual reconstructions of elongated coprolites (Krasiejów). $a_i$,** Coprolite ZPAL AbIII/3433 attributed to *Paleorhinus* which contains numerous fish remains. $a_{ii}$, virtual thin section. $a_{iii}$, Fish bone (girdle?). $a_{iv}$, Fish bone. $a_v$, Close up of fish scales. $b_i$, Specimen ZPAL AbIII/3435 in semi-transparency a few possible arthropod inclusions. $b_{ii}$, A possible arthropod fragment. $b_{iii}$, Virtual thin section. $c_i$, Specimen ZPAL AbIII/3436 in semi-transparency numerous insect remains. $c_{ii}$, Beetle elytra. $c_{iii}$, Tube with "segments" or rolled plant fragments. $c_{iv}$, A possible pectinate or lamellate beetle antenna. $c_v$-$c_{vi}$, Arthropod appendages showing resemblance to mandibles of tiger beetles (cicindelids). $c_{vii}$, Arthropod appendage. $c_{viii}$, A possible invertebrate jaw element. $d_i$, Semi-transparent coprolite ZPAL AbIII/3434 with numerous insect and plant inclusions. $d_{ii}$, One isolated fibre. $d_{iii}$, Beetle elytra. $d_{iv}$, A bundle of fibres. $e_i$-$e_{iii}$, Fragments of coprolite ZPAL AbIII/3440, attributed to *Paleorhinus*, which contains abundant fish remains and round structures. $e_{iv}$, Tooth plate. $e_v$-$e_{vi}$, Denticle-bearing fish bones. Possibly pharyngeal ossicles from gill arches. $e_{vii}$, Articulated thin bones. $e_{viii}$, Ornamented bone. $e_{ix}$, Long bone (girdle?). $e_x$, Tooth. $e_{xi}$, bivalves. $e_{xii}$, Two of the spherical structures with spiral structure and internal core. $e_{xiii}$, Bone with tube. $e_{xiv}$, Fragment of rolled plant fragment. $e_{xv}$, Fish bone. $e_{xvi}$, Articulated fish vertebrae. $e_{xvii}$, Spherical structures. $e_{xviii}$-$e_{xx}$, Virtual thin sections showing the abundant inclusions.

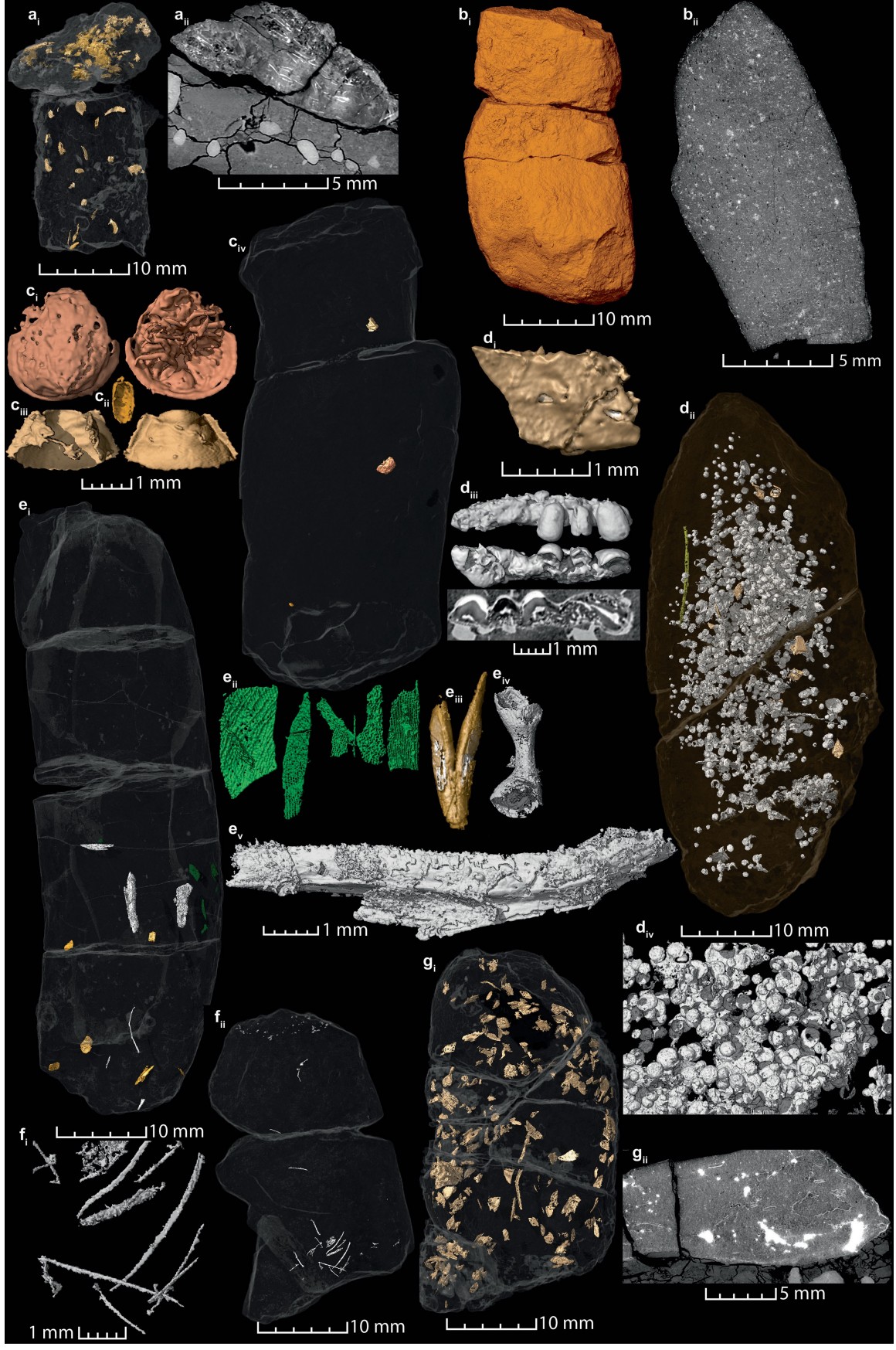

**Extended Data Fig. 4** | See next page for caption.

**Extended Data Fig. 4 | Virtual reconstructions of flat coprolites (Krasiejów).**
$a_i$, Specimen ZPAL AbIII/3505 in semi-transparency with fish inclusions.
$a_{ii}$, Virtual thin section showing the coprolite matrix and abundant fish scales.
$b_i$, Specimen ZPAL AbIII/3502 rendered in 3D. $b_{ii}$, Virtual thin section of the
coprolite matrix with numerous small, but unidentifiable inclusions. $c$, Images
from coprolite ZPAL AbIII/3504 including a crustacean (possibly a cycloid
larva) ($c_i$), a beetle elytron ($c_{ii}$), a beetle pronotum ($c_{iii}$), and the coprolite with
inclusions ($c_{iv}$). $d$, Images from coprolite ZPAL AbIII/3503 including a ganoid
fish scale ($d_i$), the coprolite with numerous round flat structures and inclusions
($d_{ii}$), a tooth-bearing fish bone ($d_{iii}$), and a close up of the round structures.
$e_i$, Specimen ZPAL AbIII/3437 with a few inclusions. $e_{ii}$, Plant fragments.
$e_{iii}$, Mid-line fish scale. $e_{iv}$, Bone from the pectoral girdle of a fish. $e_v$, Fish bone.
$f_i$, Close-up of the fibrous structures. $f_{ii}$, Coprolite ZPAL AbIII/3439 with fibrous
inclusions. $g_i$, ZPAL AbIII/3438 with abundant fish scales. $g_{ii}$, Virtual thin section.

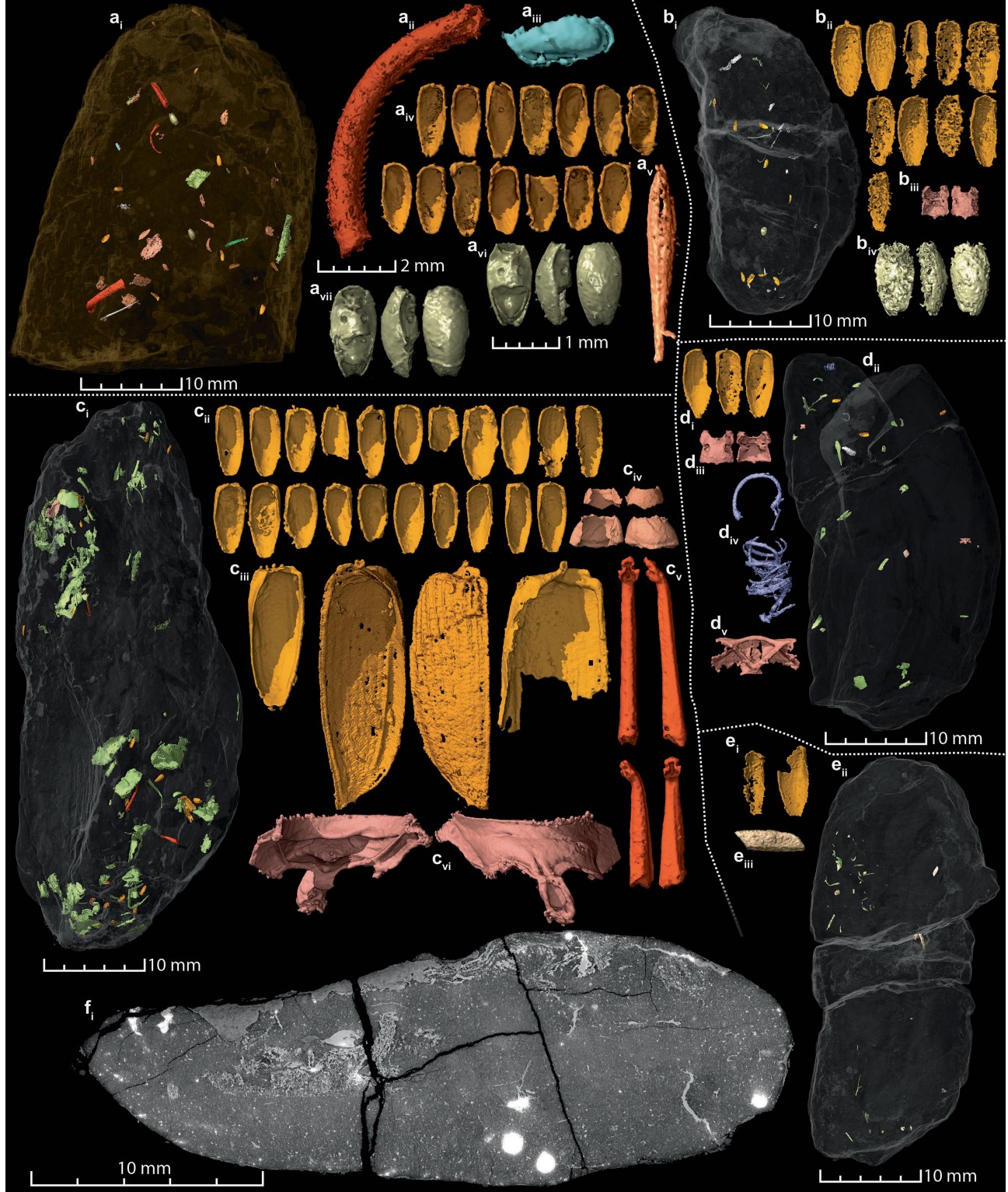

**Extended Data Fig. 5 | Coprolites assigned to *Silesaurus opolensis* (Krasiejów).** $a_i$, ZPAL AbIII/3408 with visible insect inclusions. $a_{ii}$, Curved inclusion with denticles on the concave side. $a_{iii}$, Ostracod. $a_{iv}$, Numerous beetle elytra. $a_v$, Possibly a fish scale. $a_{vi}$, Semi-articulated beetle. $a_{vii}$, Semi-articulated beetle. $b_i$, Coprolite ZPAL AbIII/3410. $b_{ii}$, Beetle elytra. $b_{iii}$, Beetle sternum. $b_{iv}$, A poorly preserved beetle body. $c_i$, ZPAL AbIII/3402. $c_{ii}$, Small elytra.

$c_{iii}$, Three large elytra. $c_{iv}$, beetle pronotums. $c_v$, two beetle tibiae of different sizes. $c_{vi}$, carabid prosternum. $d_i$, elytra. $d_{ii}$, ZPAL AbIII/3411 $d_{iii}$, Beetle sternum. $d_{iv}$, swirl-shaped inclusion, perhaps derived from some inner insect structure. $d_v$, probable insect fragment, but of unknown origin. $e_i$, ZPAL AbIII/3409. $e_{ii}$, Elytra. $e_{iii}$, Fish scale. $f_i$, Virtual thin section of ZPAL AbIII/3410 (same as in **b**).

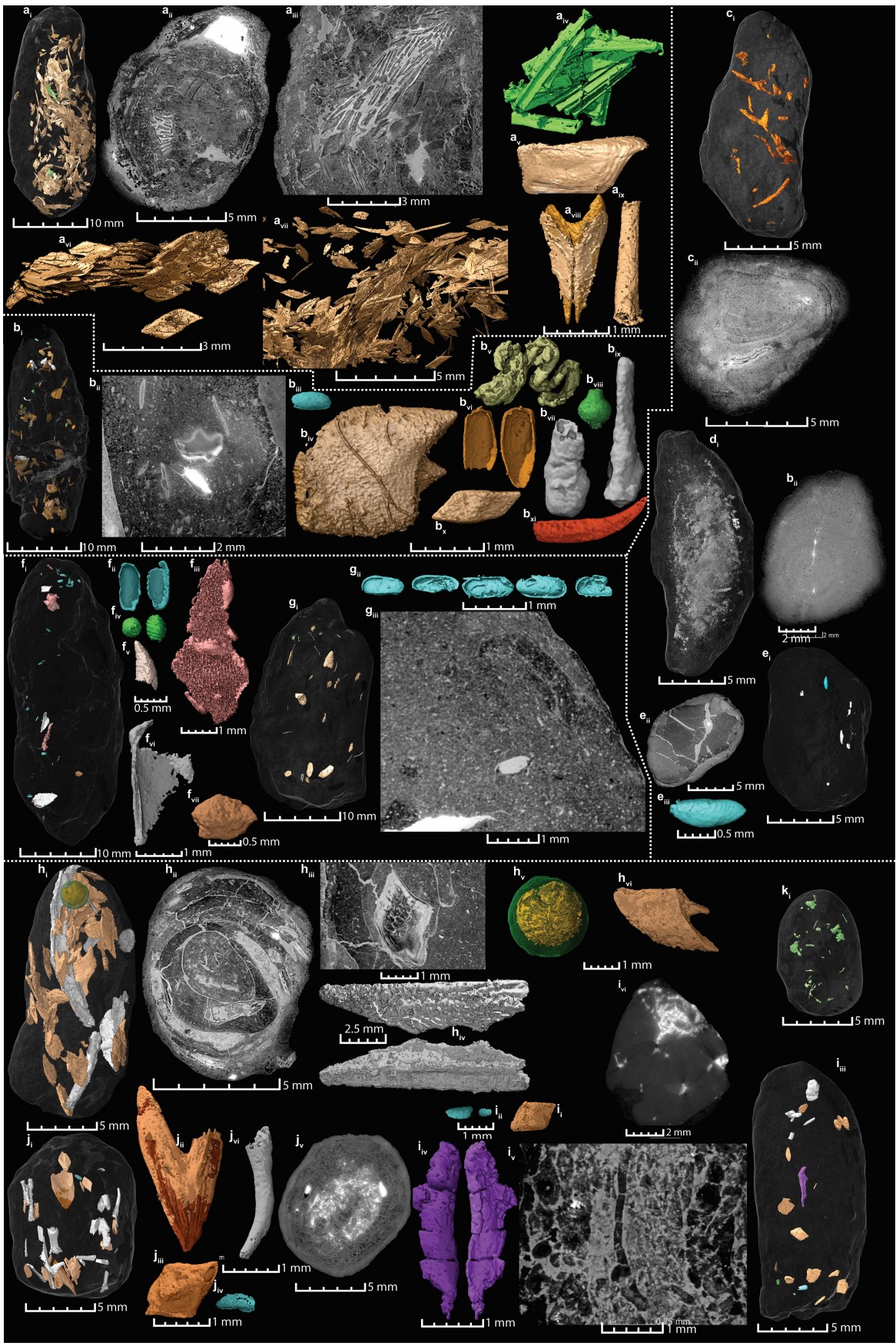

**Extended Data Fig. 6** | See next page for caption.

**Extended Data Fig. 6 | Virtual reconstructions of small elongated coprolites (a-b), small coprolites with tapered ends (c-e), small elongated coprolites (f-g), and small spiral coprolites (h-k) from Krasiejów. $a_i$,** Coprolite ZPAL AbIII/3509. $a_{ii}$-$a_{iii}$, Virtual thin sections showing the matrix and articulated scales. $a_{iv}$, Semi-articulated fin lepidotrichia. $a_v$, A close-up of a fish scale. $a_{vi}$-$a_{vii}$, Partly articulated fish scales. $a_{viii}$, midline fish scale. $a_{ix}$, Fish bone. $b_i$, Specimen ZPAL AbIII/3508 which contains fish remains and various other inclusions. $b_{ii}$, Virtual thin section. Note the fish scale in the center. $b_{iii}$, Ostracod. $b_{iv}$, Fish scale, note the peg in which the next scale would have been in articulation with. $b_v$, Enigmatic curved inclusion. $b_{vi}$, Elytra. $b_{vii}$, Fish bone. $b_{viii}$, Supposed charophyte gyrogonite. $b_{ix}$, Fish bone. $b_x$, Fish scale. $b_{xi}$, A possible pincer or mandible. $c_i$, Coprolite ZPAL AbIII/3516 with numerous burrows. $c_{ii}$, Virtual thin section. Note the burrow in the bottom centre. $d_i$, Coprolite ZPAL AbIII/3517 with dense mineralised content. $d_{ii}$, Virtual thin section. Note the numerous small inclusions. $e_i$, Coprolite ZPAL AbIII/3515. $e_{ii}$, Virtual thin section. $e_{iii}$, A possible ostracod. $f_i$, Coprolite ZPAL AbIII/3514. $f_{ii}$, Ostracod shells. $f_{iii}$, Possible arthropod cuticle. $f_{iv}$, Charophyte gyrogonite. $f_v$, Small tooth. $f_{vi}$, Fragment of a bivalve. $f_{vii}$, Fish scale. $g_i$, Coprolite ZPAL AbIII/3513 with fish remains and ostracods. $g_{ii}$, Ostracods. $g_{iii}$, Virtual thin section with an ostracod at the centre. $h_i$, Coprolite ZPAL AbIII/3511 with numerous fish remains. $h_{ii}$, Virtual thin section. Note the internal spiral structure. $h_{iii}$, Virtual thin section through a fish scale. $h_{iv}$, Ornamented bone, possibly of a fish or young temnospondyl. $h_v$, Round structure with and internal walnut-like structure. Perhaps a seed. $h_{vi}$, Fish scale. $i_i$, Fish scale. $i_{ii}$, Ostracods. $i_{iii}$, Coprolite ZPAL AbIII/3510. $i_{iv}$, Possible tapeworm proglottids. $i_v$, Virtual thin section through the possible tapeworm proglottids. $i_{vi}$, Virtual thin section. Note the internal spiral structure. $j_i$, Coprolite ZPAL AbIII/3512 with numerous fish remains. $j_{ii}$, Midline fish scale. $j_{iii}$, Lateral line fish scale. $j_{iv}$, Ostracod. $j_v$, Virtual thin section. Note the internal spiral structure. $j_{vi}$, Fish bone. $k_i$, Coprolite ZPAL AbIII/3518 with unidentified remains.

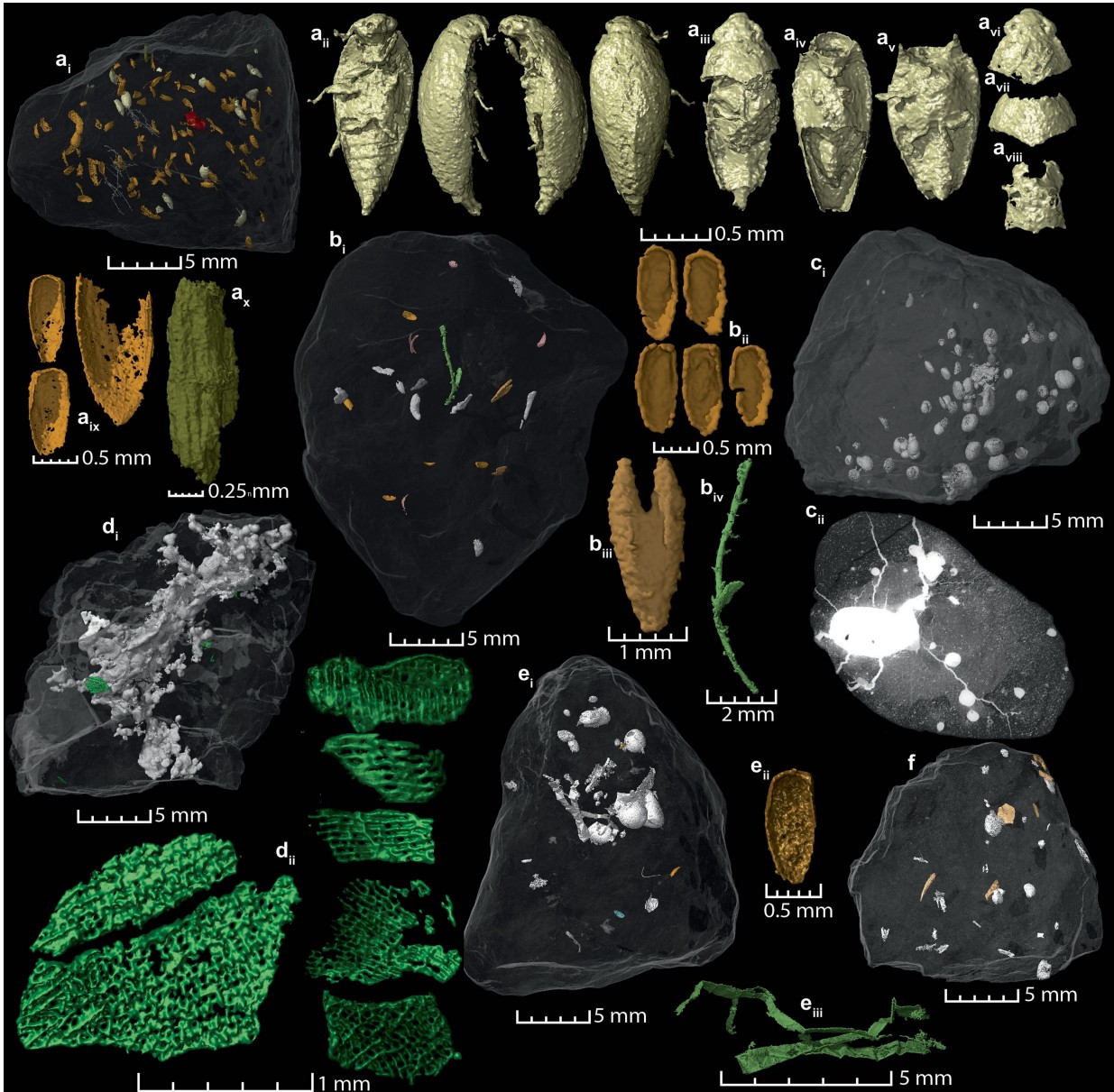

**Extended Data Fig. 7 | Fragments of coprolites (Krasiejów). $a_i$,** Coprolite fragment ZPAL AbIII/3520 that contains articulated beetles and many isolated beetle remains of *Triamyxa coprolithica* (see Qvarnström et al.[29]). $a_{ii}$, The most complete specimens. Note that fine details such as appendages and eyes are preserved. $a_{iii}$, Beetle specimen with some parts of the dorsal part of the abdomen missing. $a_{iv}$, Beetle without the head preserved. Note that the elytra are still attached. $a_v$, A slightly wider individual. $a_{vi}$, Beetle head and pronotum in articulation. $a_{vi}$, Beetle head and pronotum. $a_{vii}$, Beetle pronotum. $a_{viii}$, Ventral thorax part of a beetle. $a_{ix}$, Some of the numerous elytra. $a_x$, Possible wood fragment. $b_i$, Specimen ZPAL AbIII/3521 with remains of beetles, fish and possibly algae. $b_{ii}$, Elytra. $b_{iii}$, Midline fish scale. $b_{iv}$, Possible moss. $c_i$, Coprolite ZPAL AbIII/3526 with mineralized spherical structures. $c_{ii}$, Virtual thin section. $d_i$, Coprolite ZPAL AbIII/3523 with numerous plant remains and mineralized cracks and spherical structures. $d_{ii}$, Various plant cuticles. $e_i$, specimen ZPAL AbIII/3525. $e_{ii}$, Elytron. $e_{iii}$, Segmented elongated structure. Perhaps a plant remain or possibly flattened tapeworm proglottids. **f,** Specimen ZPAL AbIII/3519 with fish remains.

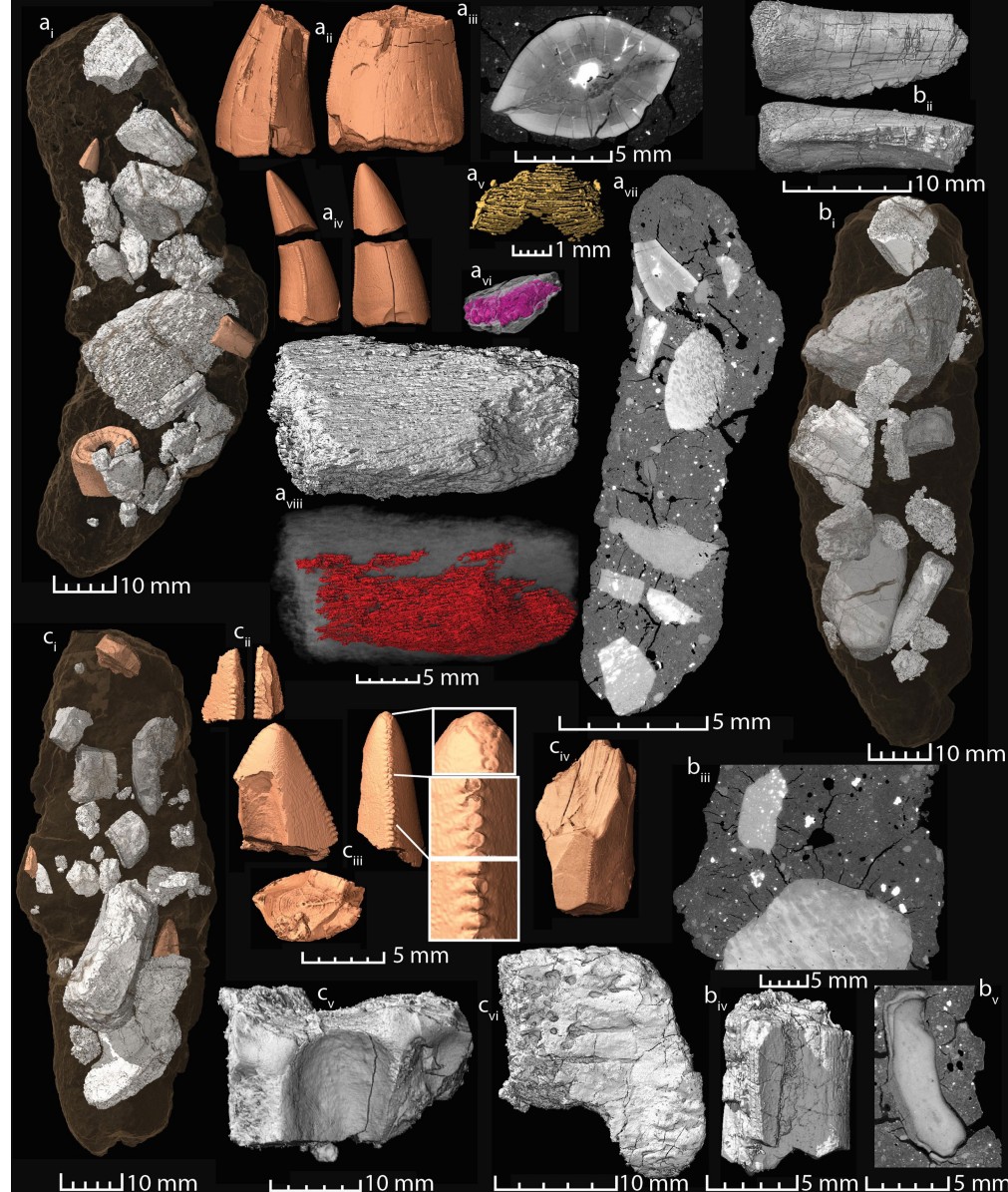

**Extended Data Fig. 8 | Coprolites assigned to *Smok wawelski* (Lisowice).**
**a_i**, Coprolite ZPAL V.33/344 with numerous bone fragments and crushed serrated teeth. **a_ii**, Worn serrated tooth. **a_iii**, A virtual thin section of the tooth in **a_ii**. **a_iv**, Two pieces of a serrated tooth that were found in different parts of the coprolite. **a_v**, A fibrous inclusion. **a_vi**, A flat bone with internal canals in oblique view. **a_viii**, virtual thin section of the entire coprolite. Note the sharp margins of the bones. **a_viii**, large bone fragment. The lower image shows the vascularization. **b_i**, ZPAL V.33/345. **b_ii**, An archosaur rib. **b_iii**, Virtual thin section. Note the cracks

in the matrix around the bones from when the dropping became dehydrated and shrunk, but the bones did not. **b_iv**, Partial rib. **b_v**, Virtual thin section of a presumed charcoal fragment. **c_i**, Coprolite ZPAL V.33/341 with numerous internal bone fragments and crushed serrated teeth. **c_ii**, A small splinter of a serrated tooth. **c_iii**, tip of a serrated tooth in various views. Note that the wear of the serrations. **c_iv**, A part of the base of a serrated tooth (probably the same tooth as c_ii and c_iii). **c_v**, Temnospondyl dermal bone. **c_vi**, Bone fragment.

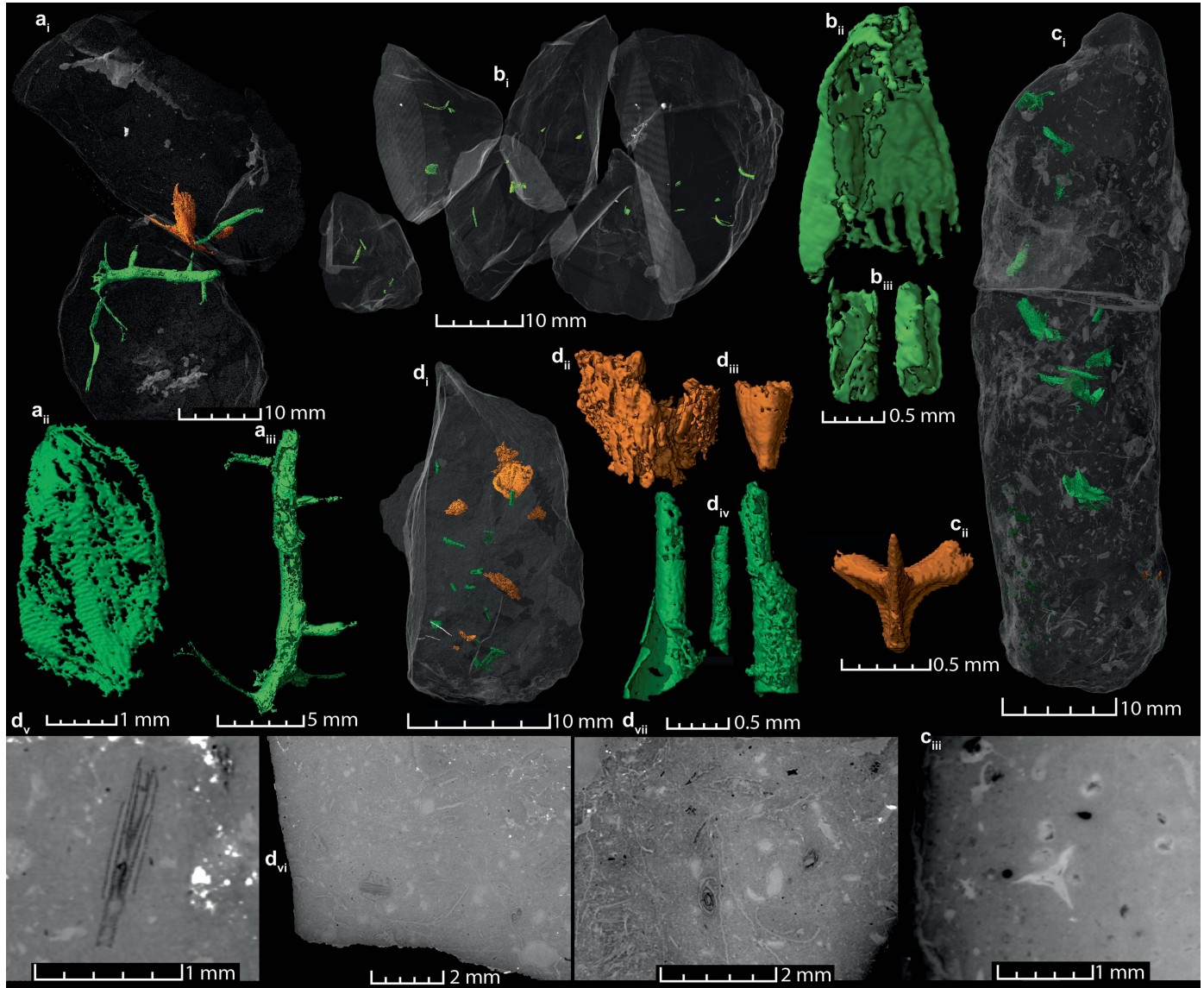

**Extended Data Fig. 9 | Plant-bearing coprolites (Lisowice). $a_i$,** Coprolite ZPAL V.33/1270 COP12 with relatively large plant fragments. $a_{ii}$, Plant fragment, which appears to be an almost complete leaf. $a_{iii}$, A branching plant fragment. $b_i$, Fragments of coprolite ZPAL V.33/1270. $b_{ii}$, Plant fragment. $b_{iii}$, Small plant (rolled fragments). $c_i$, Coprolite ZPAL V.33/1343 with relatively big plant fragments. $c_{ii}$, Thorn of a plant. $c_{iii}$, Virtual thin section of the thorn. $d_i$, Coprolite Lisowice A. $d_{ii}$, Possible wood fragment. $d_{iii}$, Plant fragment. $d_{iv}$, Various tube-shapes plant fragments. $d_v$, The structure of a tube-shapes plant fragment in a virtual thin section. $d_{vi}$, Virtual thin section showing the matrix and plant inclusions. $d_{vii}$, Various plant fragments in a virtual thin section.

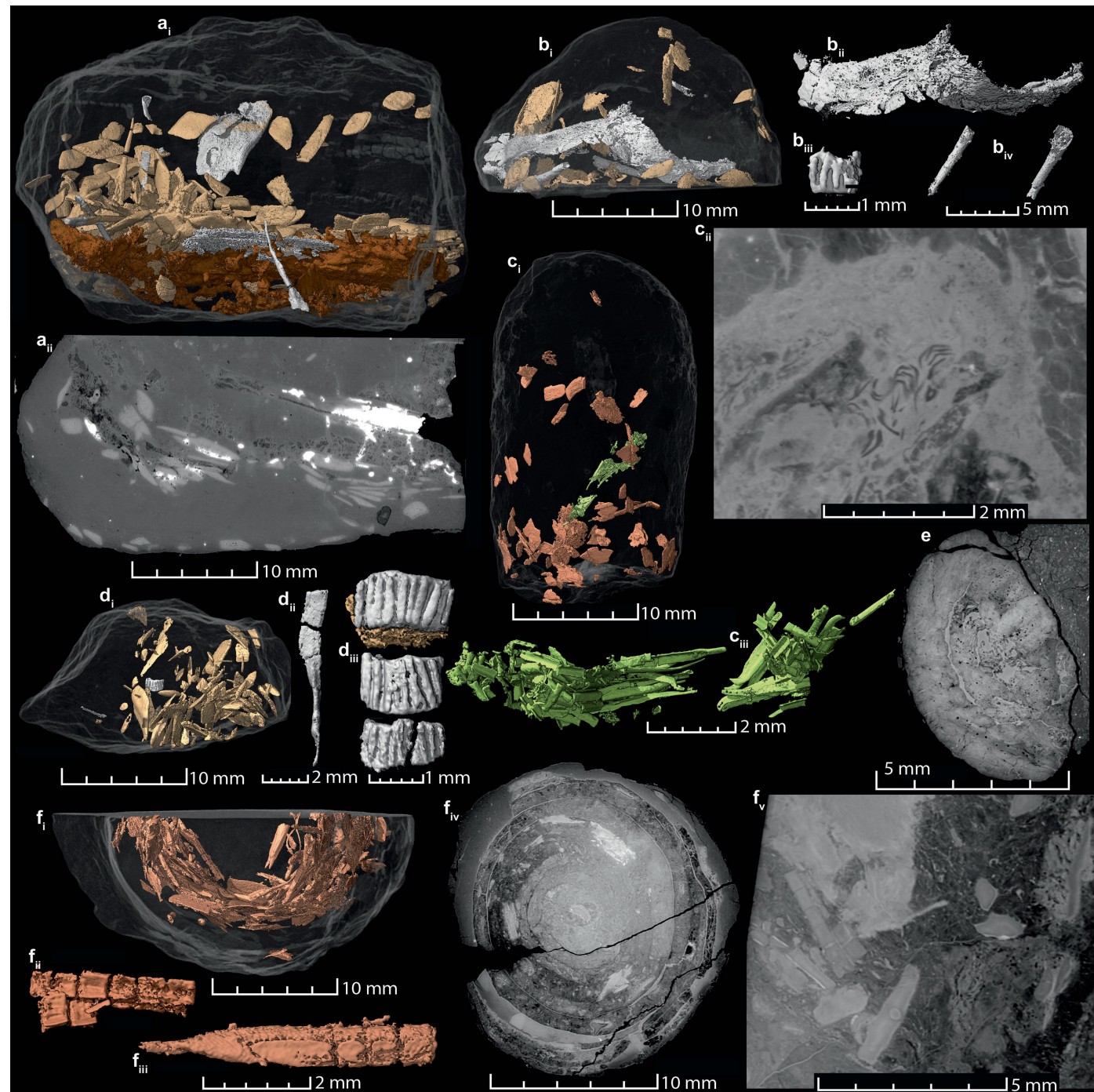

**Extended Data Fig. 10 | Fish-bearing coprolites from Lisowice. aᵢ**, Fragment of ZPAL V.33/1270 COP04 with numerous fish bones, scales and possible soft tissue (brown). **aᵢᵢ**, Virtual thin section. Note the dark area that surrounds the scales (brown structure in **aᵢ**). **bᵢ**, Fragment of V.33/1270 COP18 with numerous fish bones and scales. **bᵢᵢ**, Large bone fragment. **bᵢᵢᵢ**, Tooth plate from an actinopterygian fish. **bᵢᵥ**, Fish bones (girdle?). **cᵢ**, Coprolite ZPAL V.33/1344 with numerous scales and articulated fins. **cᵢᵢ**, Virtual thin section showing the articulated lepidotrichia in the centre. **cᵢᵢᵢ**, Articulated fin lepidotrichia.

**dᵢ**, Fragment of ZPAL V.33/1270 COP04 (as in **a**) containing numerous scales, bones and tooth plates of actinopterygian fish. **dᵢᵢ**, Fish bone. **dᵢᵢᵢ**, Tooth plates. **e**, Virtual thin section of specimen "Spiral coprolite and matrix". Note the spiral structure. **fᵢ**, Coprolite ZPAL V.33/1270 COP06 with many fish scales aligned with the spiral inner structure of the coprolite. **fᵢᵢ-fᵢᵢᵢ**, Fin lepidotrichia. **fᵢᵥ**, Virtual thin section showing the numerous spiral convolutions. **fᵥ**, Virtual thin section with a lateral line scale in the bottom centre.

# Reporting Summary

## Statistics

For all statistical analyses, confirm that the following items are present in the figure legend, table legend, main text, or Methods section.

| n/a | Confirmed | |
|---|---|---|
| ☒ | ☐ | The exact sample size (*n*) for each experimental group/condition, given as a discrete number and unit of measurement |
| ☒ | ☐ | A statement on whether measurements were taken from distinct samples or whether the same sample was measured repeatedly |
| ☒ | ☐ | The statistical test(s) used AND whether they are one- or two-sided<br>*Only common tests should be described solely by name; describe more complex techniques in the Methods section.* |
| ☒ | ☐ | A description of all covariates tested |
| ☒ | ☐ | A description of any assumptions or corrections, such as tests of normality and adjustment for multiple comparisons |
| ☒ | ☐ | A full description of the statistical parameters including central tendency (e.g. means) or other basic estimates (e.g. regression coefficient) AND variation (e.g. standard deviation) or associated estimates of uncertainty (e.g. confidence intervals) |
| ☒ | ☐ | For null hypothesis testing, the test statistic (e.g. $F$, $t$, $r$) with confidence intervals, effect sizes, degrees of freedom and $P$ value noted<br>*Give P values as exact values whenever suitable.* |
| ☐ | ☐ | For Bayesian analysis, information on the choice of priors and Markov chain Monte Carlo settings |
| ☒ | ☐ | For hierarchical and complex designs, identification of the appropriate level for tests and full reporting of outcomes |
| ☒ | ☐ | Estimates of effect sizes (e.g. Cohen's *d*, Pearson's *r*), indicating how they were calculated |

*Our web collection on statistics for biologists contains articles on many of the points above.*

## Software and code

Policy information about availability of computer code

| Data collection | Software used at beamline ID19, European Synchrotron Radiation Facility (France), NIST 17, NIS-Elements software |
|---|---|
| Data analysis | e.g. VGStudio MAX version 3.1, Illustrator/Photoshop, CorelDraw, various software associated to microscopy and geochemistry |

For manuscripts utilizing custom algorithms or software that are central to the research but not yet described in published literature, software must be made available to editors and reviewers. We strongly encourage code deposition in a community repository (e.g. GitHub). See the Nature Portfolio guidelines for submitting code & software for further information.

## Data

Policy information about availability of data

All manuscripts must include a data availability statement. This statement should provide the following information, where applicable:

- Accession codes, unique identifiers, or web links for publicly available datasets
- A description of any restrictions on data availability
- For clinical datasets or third party data, please ensure that the statement adheres to our policy

Reconstructed image stacks of the synchrotron-scanned bromalites will upon publication be publicly available in ESRF's heritage database for palaeontology, evolutionary biology and archaeology: http://paleo.esrf.eu/. Geological/palaeobotanical samples, all studied bromalite specimens, and bone with bite marks are housed in the scientific collection at the Polish Geological Institute-National Research Institute (Warszawa, Kielce; acronym Muz. PGI; Muz. PGI OS), Institute of Paleobiology, Polish Academy of Sciences (Warszawa; acronym ZPAL), in the collections of research results at the University of Silesia (Sosnowiec; palaeobotanical

data), in Paleobotanical collection Palaeozoic and Mesozoic of the National Biodiversity Collection – Herbarium KRAM at W. Szafer Institute of Botany, Polish Academy of Sciences, Cracow, Poland (KRAM) and Jagiellonian University (Kraków; palaeobotanical data).

## Research involving human participants, their data, or biological material

Policy information about studies with underline(human participants or human data). See also policy information about underline(sex, gender (identity/presentation), and sexual orientation) and underline(race, ethnicity and racism).

| Reporting on sex and gender | *Use the terms sex (biological attribute) and gender (shaped by social and cultural circumstances) carefully in order to avoid confusing both terms. Indicate if findings apply to only one sex or gender; describe whether sex and gender were considered in study design; whether sex and/or gender was determined based on self-reporting or assigned and methods used. Provide in the source data disaggregated sex and gender data, where this information has been collected, and if consent has been obtained for sharing of individual-level data; provide overall numbers in this Reporting Summary. Please state if this information has not been collected. Report sex- and gender-based analyses where performed, justify reasons for lack of sex- and gender-based analysis.* |
| --- | --- |
| Reporting on race, ethnicity, or other socially relevant groupings | *Please specify the socially constructed or socially relevant categorization variable(s) used in your manuscript and explain why they were used. Please note that such variables should not be used as proxies for other socially constructed/relevant variables (for example, race or ethnicity should not be used as a proxy for socioeconomic status). Provide clear definitions of the relevant terms used, how they were provided (by the participants/respondents, the researchers, or third parties), and the method(s) used to classify people into the different categories (e.g. self-report, census or administrative data, social media data, etc.) Please provide details about how you controlled for confounding variables in your analyses.* |
| Population characteristics | *Describe the covariate-relevant population characteristics of the human research participants (e.g. age, genotypic information, past and current diagnosis and treatment categories). If you filled out the behavioural & social sciences study design questions and have nothing to add here, write "See above."* |
| Recruitment | *Describe how participants were recruited. Outline any potential self-selection bias or other biases that may be present and how these are likely to impact results.* |
| Ethics oversight | *Identify the organization(s) that approved the study protocol.* |

Note that full information on the approval of the study protocol must also be provided in the manuscript.

## Field-specific reporting

Please select the one below that is the best fit for your research. If you are not sure, read the appropriate sections before making your selection.

☐ Life sciences        ☐ Behavioural & social sciences        ☒ Ecological, evolutionary & environmental sciences

For a reference copy of the document with all sections, see underline(nature.com/documents/nr-reporting-summary-flat.pdf)

## Ecological, evolutionary & environmental sciences study design

All studies must disclose on these points even when the disclosure is negative.

| Study description | Study of fossil specimens across the Upper Triassic to Lower Jurassic interval in the Polish Basin |
| --- | --- |
| Research sample | Hundreds of fossils with direct evidence of feeding (including coprolites (fossil droppings), regurgitalites (fossil regurgitates), and bite-marked bones) plant fossils, and geological samples. |
| Sampling strategy | All different kinds of bromalites were analysed in order to study all possible ecological interactions |
| Data collection | The bromalites derive from natural or artificial Upper Triassic and Lower Jurassic sites exposures located in Silesia and Holy Cross Mts. in the Polish Basin area (see Supp. Fig. 1). A total of 532 bromalites have been collected from eight fossiliferous sites (Supp. Tabs. 2-9). The specimens were collected between 1996 and 2017 by G.N., T.S., K.O., G.Pi., and M.Q. |
| Timing and spatial scale | Fieldwork: 1996 and 2017. Synchrotron data were collected in two scanning sessions during 2016. Data analysis collection and analyses have been ongoing since. |
| Data exclusions | No data was excluded from the analysis |
| Reproducibility | All methods are described carefully, samples are stored in appropriate collections, and imaging data will be publically available. |
| Randomization | Not relevant for this study of fossil specimens |
| Blinding | Not relevant for this study of fossil specimens |

| Did the study involve field work? | ☒ Yes | ☐ No |
|---|---|---|

## Field work, collection and transport

| Field conditions | Fieldwork was conducted during many field seasons, predominantly during summer months. |
|---|---|
| Location | Upper Triassic and Lower Jurassic sites exposures located in Silesia and Holy Cross Mts. in the Polish Basin area (see manuscript for details) |
| Access & import/export | Necessary permits were acquired from local governments for the fieldwork at the sites. |
| Disturbance | Disturbance was minimal during fieldwork |

# Reporting for specific materials, systems and methods

We require information from authors about some types of materials, experimental systems and methods used in many studies. Here, indicate whether each material, system or method listed is relevant to your study. If you are not sure if a list item applies to your research, read the appropriate section before selecting a response.

### Materials & experimental systems

| n/a | Involved in the study |
|---|---|
| ☒ | ☐ Antibodies |
| ☒ | ☐ Eukaryotic cell lines |
| ☐ | ☒ Palaeontology and archaeology |
| ☒ | ☐ Animals and other organisms |
| ☒ | ☐ Clinical data |
| ☒ | ☐ Dual use research of concern |
| ☒ | ☐ Plants |

### Methods

| n/a | Involved in the study |
|---|---|
| ☒ | ☐ ChIP-seq |
| ☒ | ☐ Flow cytometry |
| ☒ | ☐ MRI-based neuroimaging |

## Palaeontology and Archaeology

| Specimen provenance | The specimens derive from Upper Triassic and Lower Jurassic sites exposures located in Silesia and Holy Cross Mts. in the Polish Basin area (see manuscript for details). Permits from the local government were obtained for fieldwork. |
|---|---|
| Specimen deposition | Geological/palaeobotanical samples, all studied bromalite specimens, and bone with bite marks are housed in the scientific collection at the Polish Geological Institute-National Research Institute (Warszawa, Kielce; acronym Muz. PGI; Muz. PGI OS), Institute of Paleobiology, Polish Academy of Sciences (Warszawa; acronym ZPAL), in the collections of research results at the University of Silesia (Sosnowiec; palaeobotanical data), in Paleobotanical collection Palaeozoic and Mesozoic of the National Biodiversity Collection – Herbarium KRAM at W. Szafer Institute of Botany, Polish Academy of Sciences, Cracow, Poland (KRAM) and Jagiellonian University (Kraków; palaeobotanical data). |
| Dating methods | No new dating data are provided |

☐ Tick this box to confirm that the raw and calibrated dates are available in the paper or in Supplementary Information.

| Ethics oversight | *Identify the organization(s) that approved or provided guidance on the study protocol, OR state that no ethical approval or guidance was required and explain why not.* |
|---|---|

Note that full information on the approval of the study protocol must also be provided in the manuscript.

## Plants

Seed stocks

*Report on the source of all seed stocks or other plant material used. If applicable, state the seed stock centre and catalogue number. If plant specimens were collected from the field, describe the collection location, date and sampling procedures.*

Novel plant genotypes

*Describe the methods by which all novel plant genotypes were produced. This includes those generated by transgenic approaches, gene editing, chemical/radiation-based mutagenesis and hybridization. For transgenic lines, describe the transformation method, the number of independent lines analyzed and the generation upon which experiments were performed. For gene-edited lines, describe the editor used, the endogenous sequence targeted for editing, the targeting guide RNA sequence (if applicable) and how the editor was applied.*

Authentication

*Describe any authentication procedures for each seed stock used or novel genotype generated. Describe any experiments used to assess the effect of a mutation and, where applicable, how potential secondary effects (e.g. second site T-DNA insertions, mosiacism, off-target gene editing) were examined.*

