## [Peer Review File · Nature]

Manuscript Title: Digestive contents and food webs record the advent of dinosaur supremacy

Reviewer Comments & Author Rebuttals

Reviewer Reports on the Initial Version:

Referees' comments:

Referee #1 (Remarks to the Author):

I very much enjoyed reading your manuscript, "Fossil digestive contents and trophic dynamics record the advent of dinosaur supremacy." I found the approach of reconstructing trophic relationships and the methodology to be refreshingly original. In summary, I feel that your interpretation of the stepwise radiation of dinosaurs to be well supported by the data.

Overall, I find the writing to be quite clear and the manuscript well structured.

There are just a few areas that I feel could be improved, however. Firstly, in my view, more attention should be given to paleoclimate. The Discussion provides a brief overview of Late Triassic climate changes from mid-Carnian through Rhaetian, citing periods of increasing or decreasing humidity, but these are not tied specifically to changes in vegetation. And how would the CAMP eruptions have impacted the vegetation? (I note that Barbacka et al. (2017) did not find any significant floral extinctions at TJB) I suggest expanding the discussion of climate trends to include the impacts on the flora.

Continuing on this theme, I see that the sample sites exposed some paleosols that could have been a source of local paleoclimate information. I'm a little surprised nothing was done with these.

Another point that I believe would strengthen the manuscript is more global context. You mention the Central European Basin, but what about the record of dinosaur radiation in southern Pangaea? Perhaps a comparison of timing in the Polish BASIN, or Northern Hemisphere in general, with the Southern Hemisphere, or maybe just the Ischigualasto Basin.

One very specific comment - what is the source of the pCO₂ data in Figure 1A? There are multiple proxies for pCO₂, and they do not always concur. Perhaps you might modify the figure to illustrate both paleosol and stomatal based proxy data.

Referee #2 (Remarks to the Author):

This study investigates trophic dynamics of dinosaurs during the Late Triassic-earliest Jurassic. The authors examine bromolites from five vertebrate assemblages of the Polish Basin in central Europe, providing a well-constrained case study for the food webs that existed as dinosaurs rose to dominance. The findings document how the bromolites increase in both size and diversity across the study interval, pointing to the emergence of larger dinosaur faunas with novel feeding patterns. When integrated with climate, these results suggest stepwise changes in dinosaur dietary guilds as the climate changes across this interval.

This study represents an important addition to the literature on early dinosaur evolution. The work presented in the manuscript is robust and meticulous, and clearly highlights the Polish Basin as a key locality in understanding the rise of dinosaurs. The authors have evidently conducted exceptionally high-quality work in this regard. More broadly, the use of bromolites (of which excellent specimens are extremely rare in the fossil record, let alone in the record of early dinosaurs) represents a unique and novel approach to understanding the dietary guilds of the earliest dinosaurs. More specifically, I find the approaches and methodologies used here to be appropriate, and the results from these to be presented accurately and appropriately.

My primary concern, as a specialist in dinosaur macroevolution, is the framing of the paper in relation to recent work on the influence of climate on early dinosaur evolution. The detailed results from the bromolites are integrated with information on climate change across the Late Triassic-earliest Jurassic interval, but there is no reference to the two most recent papers that link dinosaur diversification to climatic events, namely: Dunne et al. (2023; <https://doi.org/10.1016/j.cub.2022.11.064>) and Griffin et al. (2022; <https://doi.org/10.1038/s41586-022-05133-x>). These papers, alongside Kent & Clemmensen (2022), which is cited in the manuscript, demonstrate how climate conditions suppressed the global distribution and diversification of early dinosaurs.

This study adds even further weight to the climate-hypothesis of early dinosaur evolution outlined in recent papers, while also adding critical information on diets and how this relates to the surrounding environment. I don't believe the current framing to be a flaw, but rather, recommend that the manuscript be reframed to directly connect with the findings of Dunne, Griffin, and colleagues. This would greatly strengthen the manuscript and place it firmly within the body of work that will become seminal research on the early evolution of the dinosaurs.

Below, I provide a few minor comments that would enhance the clarity of the manuscript:

Line 30: "...stepwise rise of dinosaurs within the Polish Basin." - Is it possible to be more specific here? Stepwise in terms of dietary guild or trophic interactions?

Line 56-57: This paragraph of the introductory section should contain a reference to the previous work on the influence of climate on dinosaur evolution (see above)

Line 65: "...making our study one of the first of its kind." - this is already apparent from the unique

approach and detailed work on the bromolites etc. It doesn't need to be stated explicitly

Line 71: This paragraph would be better placed in second-to-last place or earlier as it provides important context on the significance of the wonderful Polish sites.

Line 84: "show a big disparity" - suggested rewording = "show great disparity"

Lines 254-259: Lacking references to recent work on the impact of climate on early dinosaur diversity

Line 263: "...experienced the heaviest blow" - in general, I prefer this (fun) wording, but it would be great to be more specific (e.g. "greatest reduction in diversity" etc.)

Line 263: "The disappearance of these formerly dominant tetrapods is mirrored by an increased abundance of dinosaurs in the body and trace fossil record" - this finding is also documented in previous papers, please refer to at least one of: Wang & Dodson 2006; Starrfelt & Liow 2016; Bernardi et al. 2018; Dunne et al. 2021)

Line 266: The five phases should be mentioned in the abstract in some capacity as this is a key finding in my opinion

Lines 276-278: "the resolution of the obtained dataset does not allow us to assess..." - I recommend rewording and reframing this to be more positive. The present dataset might not be able to do exactly this, but what an exciting opportunity for future work!

Referee #3 (Remarks to the Author):

Dear Editor,

Please find below the review for the paper “Fossil digestive contents and trophic dynamics record the advent of dinosaur supremacy” by Qvarnström and coauthors.

This is an excellent, meticulously documented, and well-illustrated paper that makes significant contributions to our understanding of the association and fossil material under study, documenting a pivotal event in dinosaur macroevolution. The paper is written in a clear and accessible language, formulating well-defined hypotheses and transparently reporting limitations. I commend the authors for their rigorous work and detailed documentation of their material, it is refreshing to see such a high quality work on dinosaur palaeontology in *Nature*, backing up analytical work with classic, primary data (stratigraphy and fossils).

I have a few minor suggestions for revision:

1. Lines 226-228: When referencing n-alkanes, it would be helpful to include a more specific reference to the Supplementary Information (SI) material. This will provide readers with direct access to additional data and context which cannot be placed in the main text for space constraints, but is highly valuable (and very well presented in the SI).

2. Lines 253: The discussion on the turnover in herbivore communities, specifically the replacement by ornithischians and sauropodomorphs, is intriguing. The authors suggest that sauropodomorph dispersal might have been influenced by the attenuation of climatic barriers. As someone interested in the macroevolutionary responses of dinosaurs to climate, I recommend that the authors expand on this topic (no more than two lines are needed), comparing and discussing these results in light of recent studies such as Langer and Godoy (2022) and Chiarenza et al. (2024). This will enhance the depth of the discussion and provide a contemporary context.

References:

M.C. Langer, P.L. Godoy. 2022. So volcanoes created the dinosaurs? A quantitative characterization of the early evolution of terrestrial Pan-Aves. *Front. Earth Sci.*, 10 (2022), <https://doi.org/10.3389/feart.2022.899562>

Chiarenza A. A., Cantalapiedra J. L., Jones L. A., Gamboa S., Galván S., Farnsworth A. J., Valdes P. J., Sotelo G., Varela S. 2024. Early Jurassic origin of avian endothermy and thermophysiological diversity in dinosaurs. *Current Biology*. 34 (11): 1–11. DOI: <https://doi.org/10.1016/j.cub.2024.04.051>

3. Line 291: The identification of five distinct phases in the dinosaur turnover process is significant. While these phases are likely illustrated in Figure 1, I suggest adding explicit statements in the text (perhaps in the form of bullet points) immediately following this sentence. Additionally, a clearer reference to Figure 1 would enhance accessibility and comprehension for the reader.

Despite these minor suggestions, I believe the paper is of high quality and will make a valuable

contribution to the field. I congratulate the authors on their work and look forward to seeing this paper published in Nature.

Best regards.

Referee #4 (Remarks to the Author):

This study examines bromalites (coprolites and other trace fossils from feeding activities) from eight Upper Triassic to Lower Jurassic sites in Poland to examine changes in trophic dynamics as dinosaurs rapidly diversified and replaced many non-dinosaurian vertebrates. The authors report that changes in both bromalites and dinosaur diversity occur in conjunction with paleoenvironmental changes during the end of the Triassic. In particular, the study demonstrates that increases in bromalite size and diversity are concurrent with increasing dinosaur diversity, body sizes, and the apparent expansion of dinosaur diets. The authors suggest that paleoenvironmental events facilitated trophic changes and that “dinosaurs rose to supremacy in a step-wise fashion across 30 million years of evolution”. As such, the study effectively uses multiple types of evidence to shed light on dinosaurs within Late Triassic food webs. This is an innovative and ambitious undertaking that integrates considerable body and trace fossil data. Nevertheless, there are some ambiguities and omissions that muddy the significance of the interpretations. I think the manuscript would be considerably strengthened if the authors address the following issues.

- The introduction to this paper describes different models that have attempted to explain the radiation of dinosaurs. This research question helps explain why this study was undertaken and the significance of the research. However, the Discussion does not address how the results of the study inform the ongoing arguments about the rapid evolutionary success of the dinosaurs. Although macroevolutionary topics are not the main focus of this study, the paper should discuss whether the documented changes in trophic patterns from the Late Triassic to the Early Jurassic might shed light on macroevolutionary mechanisms. Some questions to consider: Why didn't the vegetation changes and availability of larger herbivore prey open up ecospace for both existing non-dinosaurian herbivores and carnivores as well as dinosaurs? Is it likely that anatomical differences, phenotypic plasticity, competition, and/or other factors played major roles in changes in trophic patterns, the extinction of many non-dinosaurian vertebrates, and the radiation of the dinosaurs?

It would also be useful if the authors explain what they mean by “dinosaur-dominated terrestrial ecosystems” (line 44) or that “dinosaurs rose to supremacy” (line 289)? Does this refer to: a) the loss of numerous non-dinosaurian vertebrates as dinosaurs diversified? b) the numerical dominance of dinosaurs relative to other vertebrates? c) the size or diurnal dominance of dinosaurs – or other factors? It is difficult to understand how different factors contribute to dinosaur evolution if “domination” or “supremacy” are not defined.

- The large sample size of bromalites in this study offers solid evidence for vertebrate feeding activity, however the authors should acknowledge the important role of taphonomic biases in interpreting the bromalite record. Feces and other digestive residues deposited in certain habitats (e.g., aquatic) are more likely to be preserved than those deposited in areas with low rates of

sedimentation. It would be informative to note whether the different localities represent similar depositional environments. Moreover, feces from carnivores generally have a higher preservation potential than herbivore feces. As such, it is likely that the bromalite records of herbivore feeding habits are less well-represented.

The paper does mention the taphonomic implications of organic geochemical analyses of some of the bromalites, but this is a limited discussion of a subset of specimens. The possible biases of taphonomy conditions certainly does not lessen the informative value of this research; this paper presents a creative approach to examining ecosystem evolution. However, the authors should acknowledge the variability of bromalite preservation potential and consider how it might skew interpretations.

- This manuscript is unique because it emphasizes the utility of information derived from >500 bromalites from the Polish Basin. Yet, a synopsis of detailed information about the bromalite assemblages is difficult to find in the text. Figure 1 offers some information about how the bromalite assemblages compare with each other (although much of the type is unreadable and there is no explanation for numbers in the “Trophic structures” section of the figure). In addition, the supplemental material offers a detailed spreadsheet of the individual bromalites studied. Nevertheless, the paper would be much more informative if it included a table describing key characteristics of each bromalite assemblage. How many (percentages from each locality) bromalites contain specific dietary inclusions such as plants, arthropods, fish, or tetrapod parts? How many specimens in a given assemblage were studied with CT scans, thin sections, or surface analyses? What was the sedimentology/depositional environment of the site the assemblage was found in? How many of the bromalites in each assemblage are coprolites? This last question is relevant because gut contents provide different information from coprolites. Indeed, it is not clear from the text or supplements whether any of the bromalites from the Polish Basin appear to be gut contents or regurgitalites. (If all of the bromalites are coprolites, why not say this?) Since the manuscript stresses the unique approach of analyzing bromalites, I suggest that the authors lean into this innovative aspect of the paper by emphasizing the informative value of bromalites and providing comprehensive information about differences between the bromalite assemblages in a table or figure. This would help readers better understand how the bromalite assemblages changed through time and how these differences relate to changes in the vertebrate fauna.

Synopsis:

I thoroughly appreciate the integration of bromalite and body fossil evidence in documenting changes in trophic patterns from the Late Triassic to the Early Jurassic in this study; it is a refreshing approach to understanding the evolution of ancient dinosaur communities. However, I suggest that the authors strengthen the manuscript by: 1) directly relating the research results to the overarching question of why dinosaur taxa and populations increased at the same time that many non-dinosaurian vertebrates lost diversity or went extinct, 2) acknowledging that taphonomic factors often contribute to a biased fossil record of fossilized digestive residues, and 3) more strongly emphasizing the broader implications of this research.

Miscellaneous questions:

Line 222:

What does the “one-sided content of Late Triassic herbivore bromalites” mean?

Line 245:

Couldn't “reconfigurations of the floral assemblages in the region” have had a positive effect on some herbivores instead of having “knock-down effects on the tetrapod communities”?

Author Rebuttals to Initial Comments:

Answers to the review comments:

Below is a detailed account of the changes made in response to the reviewers.

Referee #1 (Remarks to the Author):

I very much enjoyed reading your manuscript, "Fossil digestive contents and trophic dynamics record the advent of dinosaur supremacy." I found the approach of reconstructing trophic relationships and the methodology to be refreshingly original. In summary, I feel that your interpretation of the stepwise radiation of dinosaurs to be well supported by the data.

Overall, I find the writing to be quite clear and the manuscript well structured.

There are just a few areas that I feel could be improved, however. Firstly, in my view, more attention should be given to paleoclimate. The Discussion provides a brief overview of Late Triassic climate changes from mid-Carnian through Rhaetian, citing periods of increasing or decreasing humidity, but these are not tied specifically to changes in vegetation. And how would the CAMP eruptions have impacted the vegetation? (I note that Barbacka et al. (2017) did not find any significant floral extinctions at TJB) I suggest expanding the discussion of climate trends to include the impacts on the flora.

Thank you for this important comment. We have added a few additional sentences to the main text, which supplement the climatic/floral information in Fig. 1 (it is mainly a summary of data on the record of CAMP in the Polish Basin presented by Pieńkowski et al. 2012, 2014). However, we must emphasize that at this stage we cannot clearly determine the more detailed scenario of floral changes and climatic fluctuations in the Late Triassic. Palynological studies indicate three major humid episodes in Late Triassic of the Polish Basin, but the plant macrofossil finds are quite scattered and do not allow for detailed comparisons, but are luckily associated with vertebrate finds. Similarly, with the paleoclimate record, it is clear that we have a change from an arid climate to a humid climate in the Carnian-Rhaetian. This is clearly reflected in the record of depositional environments, clay mineralogy, and palaeosols characters (general trend is illustrated in Fig. 1). The Late Triassic flora of the Polish Basin is still being studied, and the preliminary results of research on flora remains preserved in bromalites are surprising (e.g. Barbacka et al. 2022; data presented in this manuscript), which well illustrates the complexity of the issue. At this stage, we can show a general picture of floristic changes in the Late Triassic (three floral macrofossil assemblages; three plant microfossil zones). The results presented by Pieńkowski et al. (2012, 2014) allow for more detailed palynological interpretations at Triassic-Jurassic boundary in the Polish Basin (summarized at Fig. 1).

We plan to develop this topic of paleoclimate and floral changes in the future, but it requires several years of studies on floras.

Continuing on this theme, I see that the sample sites exposed some paleosols that could have been a source of local paleoclimate information. I'm a little surprised nothing was done with these.

We agree with the referee that there is potential to study the area's paleoclimate using palaeosols, and in fact, this is part of ongoing research. However, more work is needed to collect and interpret these data, and it is beyond the scope of this paper. A few years ago, we conducted a pilot study of palaeosols from the Rhaetian and late Norian (boreholes from Pomerania in N Poland and exposures in the south, Silesia/Holy Cross Mts.). There is a great contrast between them: 1) late Norian palaeosols are rich in carbonates and show soil-forming processes typical of the semi-arid climate. Rhaetian sediments lack carbonate-rich palaeosols, and pedogenic horizons are enriched in rare earth elements. So, there's an interesting topic there to dig into.

Another point that I believe would strengthen the manuscript is more global context. You mention the Central European Basin, but what about the record of dinosaur radiation in southern Pangaea? Perhaps a comparison of timing in the Polish BASIN, or Northern Hemisphere in general, with the Southern Hemisphere, or maybe just the Ischigualasto Basin.

We agree with the reviewer that comparing the record from the Polish Basin with that of the Southern Hemisphere would be very interesting and is a fascinating area for future research (requiring extensive data synthesis). We have not expanded the study to an even broader global context because of stratigraphical uncertainties and the complexity of this evolutionary event. We believe that focusing on an area where the event is recorded in high resolution within a controlled stratigraphy is essential to unravel the nature of the event. We agree with the reviewer that this would be the natural next step of the study: comparing our data with that of different areas (other parts of the Central European Basin, the Jameson Land Basin, E Greenland, and more southern basins) to further unravel the complex nature of early dinosaur evolution.

One very specific comment - what is the source of the pCO₂ data in Figure 1A? There are multiple proxies for pCO₂, and they do not always concur. Perhaps you might modify the figure to illustrate both paleosol and stomatal based proxy data.

The Pco₂ curve is based on soil carbonate proxies, which is now clarified in the figure caption along with a reference to Olsen et al., 2022.

Referee #2 (Remarks to the Author):

This study investigates trophic dynamics of dinosaurs during the Late Triassic-earliest

Jurassic. The authors examine bromalites from five vertebrate assemblages of the Polish Basin in central Europe, providing a well-constrained case study for the food webs that existed as dinosaurs rose to dominance. The findings document how the bromolites increase in both size and diversity across the study interval, pointing to the emergence of larger dinosaur faunas with novel feeding patterns. When integrated with climate, these results suggest stepwise changes in dinosaur dietary guilds as the climate changes across this interval.

This study represents an important addition to the literature on early dinosaur evolution. The work presented in the manuscript is robust and meticulous, and clearly highlights the Polish Basin as a key locality in understanding the rise of dinosaurs. The authors have evidently conducted exceptionally high-quality work in this regard. More broadly, the use of bromolites (of which excellent specimens are extremely rare in the fossil record, let alone in the record of early dinosaurs) represents a unique and novel approach to understanding the dietary guilds of the earliest dinosaurs. More specifically, I find the approaches and methodologies used here to be appropriate, and the results from these to be presented accurately and appropriately.

My primary concern, as a specialist in dinosaur macroevolution, is the framing of the paper in relation to recent work on the influence of climate on early dinosaur evolution. The detailed results from the bromolites are integrated with information on climate change across the Late Triassic-earliest Jurassic interval, but there is no reference to the two most recent papers that link dinosaur diversification to climatic events, namely: Dunne et al. (2023; <https://doi.org/10.1016/j.cub.2022.11.064>) and Griffin et al. (2022; <https://doi.org/10.1038/s41586-022-05133-x>). These papers, alongside Kent & Clemmensen (2022), which is cited in the manuscript, demonstrate how climate conditions suppressed the global distribution and diversification of early dinosaurs.

This study adds even further weight to the climate-hypothesis of early dinosaur evolution outlined in recent papers, while also adding critical information on diets and how this relates to the surrounding environment. I don't believe the current framing to be a flaw, but rather, recommend that the manuscript be reframed to directly connect with the findings of Dunne, Griffin, and colleagues. This would greatly strengthen the manuscript and place it firmly within the body of work that will become seminal research on the early evolution of the dinosaurs."

Following the suggestion from reviewer #2, we have now cited these works, along with two additional studies suggested by reviewer #3. Now, the manuscript has a better anchoring in the recent literature on how the climate affected the distribution and diversification of early dinosaurs.

Below, I provide a few minor comments that would enhance the clarity of the manuscript:

Line 30: "...stepwise rise of dinosaurs within the Polish Basin." - Is it possible to be more specific here? Stepwise in terms of dietary guild or trophic interactions?

This is rephrased to "...stepwise increase of dinosaur diversity and ecospace occupancy..." for clarification.

Line 56-57: This paragraph of the introductory section should contain a reference to the previous work on the influence of climate on dinosaur evolution (see above)

These references are now cited.

Line 65: "...making our study one of the first of its kind." - this is already apparent from the unique approach and detailed work on the bromolites etc. It doesn't need to be stated explicitly

This sentence was removed following the suggestion of the reviewer.

Line 71: This paragraph would be better placed in second-to-last place or earlier as it provides important context on the significance of the wonderful Polish sites.

The last two paragraphs have been exchanged and rephrased to enhance clarity and flow.

Line 84: "show a big disparity" - suggested rewording = "show great disparity"
rephrased.

Lines 254-259: Lacking references to recent work on the impact of climate on early dinosaur diversity

The mentioned references are now cited in this paragraph.

Line 263: "...experienced the heaviest blow" - in general, I prefer this (fun) wording, but it would be great to be more specific (e.g. "greatest reduction in diversity" etc.)

This is now rephrased following the suggestion of the reviewer.

Line 263: "The disappearance of these formerly dominant tetrapods is mirrored by an increased abundance of dinosaurs in the body and trace fossil record" - this finding is also documented in previous papers, please refer to at least one of: Wang & Dodson 2006; Starrfelt & Liow 2016; Bernardi et al. 2018; Dunne et al. 2021)

Wang & Dodson 2006 is now cited here.

Line 266: The five phases should be mentioned in the abstract in some capacity as this is a key finding in my opinion

The five phases are now highlighted by numbers in the abstract for clarification.

Lines 276-278: "the resolution of the obtained dataset does not allow us to assess..." - I recommend rewording and reframing this to be more positive. The present dataset might not be able to do exactly this, but what an exciting opportunity for future work!

We agree with the reviewer and have now rephrased this sentence. This part is also slightly expanded following the comments of other reviewers.

Referee #3 (Remarks to the Author):

Dear Editor,

Please find below the review for the paper "Fossil digestive contents and trophic dynamics record the advent of dinosaur supremacy" by Qvarnström and coauthors. This is an excellent, meticulously documented, and well-illustrated paper that makes significant contributions to our understanding of the association and fossil material under study, documenting a pivotal event in dinosaur macroevolution. The paper is written in a clear and accessible language, formulating well-defined hypotheses and transparently reporting limitations. I commend the authors for their rigorous work and detailed documentation of their material, it is refreshing to see such a high quality work on dinosaur palaeontology in Nature, backing up analytical work with classic, primary data (stratigraphy and fossils).

I have a few minor suggestions for revision:

1. Lines 226-228: When referencing n-alkanes, it would be helpful to include a more specific reference to the Supplementary Information (SI) material. This will provide readers with direct access to additional data and context which cannot be placed in the main text for space constraints, but is highly valuable (and very well presented in the SI). **A specific reference to the relevant section in the supplementary information is now included following the suggestion of the reviewer.**

2. Lines 253: The discussion on the turnover in herbivore communities, specifically the replacement by ornithischians and sauropodomorphs, is intriguing. The authors suggest that sauropodomorph dispersal might have been influenced by the attenuation of climatic barriers. As someone interested in the macroevolutionary responses of dinosaurs to climate, I recommend that the authors expand on this topic (no more than two lines are needed), comparing and discussing these results in light of recent studies such as Langer and Godoy (2022) and Chiarenza et al. (2024). This will enhance the depth of the discussion and provide a contemporary context.

References:

M.C. Langer, P.L. Godoy. 2022. So volcanoes created the dinosaurs? A quantitative characterization of the early evolution of terrestrial Pan-Aves. *Front. Earth Sci.*, 10 (2022), <https://doi.org/10.3389/feart.2022.899562>

Chiarenza A. A., Cantalapiedra J. L., Jones L. A., Gamboa S., Galván S., Farnsworth A. J., Valdes P. J., Sotelo G., Varela S. 2024. Early Jurassic origin of avian endothermy and thermophysiological diversity in dinosaurs. *Current Biology*. 34 (11): 1–11.

DOI: <https://doi.org/10.1016/j.cub.2024.04.051>

This is now more elaborated on in the text, and the two mentioned key references are included along with two additional references suggested by reviewer #2.

3. Line 291: The identification of five distinct phases in the dinosaur turnover process is significant. While these phases are likely illustrated in Figure 1, I suggest adding explicit statements in the text (perhaps in the form of bullet points) immediately following this sentence. Additionally, a clearer reference to Figure 1 would enhance accessibility and comprehension for the reader.

A reference to Figure 3b is now included following the suggestion of the reviewer.

Despite these minor suggestions, I believe the paper is of high quality and will make a valuable contribution to the field. I congratulate the authors on their work and look forward to seeing this paper published in Nature.

Best regards.

Referee #4 (Remarks to the Author):

This study examines bromalites (coprolites and other trace fossils from feeding activities) from eight Upper Triassic to Lower Jurassic sites in Poland to examine changes in trophic dynamics as dinosaurs rapidly diversified and replaced many non-dinosaurian vertebrates. The authors report that changes in both bromalites and dinosaur diversity occur in conjunction with paleoenvironmental changes during the end of the Triassic. In particular, the study demonstrates that increases in bromalite size and diversity are concurrent with increasing dinosaur diversity, body sizes, and the apparent expansion of dinosaur diets. The authors suggest that paleoenvironmental events facilitated trophic changes and that "dinosaurs rose to supremacy in a step-wise fashion across 30 million years of evolution". As such, the study effectively uses multiple types of evidence to shed light on dinosaurs within Late Triassic food webs. This is an innovative and ambitious undertaking that integrates considerable body and trace fossil data. Nevertheless, there are some ambiguities and omissions that muddy the significance of the interpretations. I think the manuscript would be considerably strengthened if the authors address the following issues.

- The introduction to this paper describes different models that have attempted to explain the radiation of dinosaurs. This research question helps explain why this study was undertaken and the significance of the research. However, the Discussion does not address how the results of the study inform the ongoing arguments about the rapid evolutionary success of the dinosaurs. Although macroevolutionary topics are not the main focus of this study, the paper should discuss whether the documented changes in trophic patterns from the Late Triassic to the Early Jurassic might shed light on

macroevolutionary mechanisms. Some questions to consider: Why didn't the vegetation changes and availability of larger herbivore prey open up ecospace for both existing non-dinosaurian herbivores and carnivores as well as dinosaurs? Is it likely that anatomical differences, phenotypic plasticity, competition, and/or other factors played major roles in changes in trophic patterns, the extinction of many non-dinosaurian vertebrates, and the radiation of the dinosaurs?

These are good points, and a discussion that we believe and hope that our paper can spark. We have now extended this part of the discussion in the manuscript.

It would also be useful if the authors explain what they mean by "dinosaur-dominated terrestrial ecosystems" (line 44) or that "dinosaurs rose to supremacy" (line 289)? Does this refer to: a) the loss of numerous non-dinosaurian vertebrates as dinosaurs diversified? b) the numerical dominance of dinosaurs relative to other vertebrates? c) the size or diurnal dominance of dinosaurs – or other factors? It is difficult to understand how different factors contribute to dinosaur evolution if "domination" or "supremacy" are not defined.

"Dinosaur-dominated ecosystems" refer to the faunas of terrestrial ecosystems of the Early Jurassic to End-Cretaceous, mainly composed of dinosaurs of various clades and trophic levels. "The rise to dinosaur supremacy" refers to the transition to this world from a former Late Triassic time when terrestrial ecosystems were composed of some early dinosaurs but also various terrestrial tetrapods (e.g. pseudosuchians, temnospondyls, rhynchosaurs and phytosaurs). We have clarified this in the two indicated sentences and throughout the manuscript as a whole.

- The large sample size of bromalites in this study offers solid evidence for vertebrate feeding activity, however the authors should acknowledge the important role of taphonomic biases in interpreting the bromalite record. Feces and other digestive residues deposited in certain habitats (e.g., aquatic) are more likely to be preserved than those deposited in areas with low rates of sedimentation. It would be informative to note whether the different localities represent similar depositional environments. Moreover, feces from carnivores generally have a higher preservation potential than herbivore feces. As such, it is likely that the bromalite records of herbivore feeding habits are less well-represented.

The paper does mention the taphonomic implications of organic geochemical analyses of some of the bromalites, but this is a limited discussion of a subset of specimens. The possible biases of taphonomy conditions certainly does not lessen the informative value of this research; this paper presents a creative approach to examining ecosystem evolution. However, the authors should acknowledge the variability of bromalite preservation potential and consider how it might skew interpretations.

It is quite remarkable how well the bromalite diversity matches the body and track fossil records of the sites.

However, as is always the case with the fossil record, we are limited to windows into the past, and there is never such a thing as a complete picture of past ecosystems, which we have not argued (our reconstructions of these ecosystems are not complete regarding the bromalite and other fossil records). As the reviewer stated, the uniqueness of the data is that it offers “solid evidence for vertebrate feeding activity.”

The depositional environments and bromalite preservation are slightly different between the sites (more data is added in an extra table). We agree with the reviewer that the main text does not discuss this much (due to limited space), but the sites are described in more detail in the SI.

A more general discussion on bromalite preservation/taphonomy is not included in the present study (although we have now included a table that provides additional information on sedimentology and taphonomy) but can be found elsewhere (e.g. Qvarnström et al., 2016) and on-going research (see comments below). Indeed, the literature often states that phosphatic bromalites from carnivores have better preservation potential than carbonaceous herbivore bromalites (although herbivore dung should have been more plentifully produced than carnivore scat). However, this can partly be due to a recognition bias—accumulation of size-selected plant matter (in nodules or just as patches in the rock) could be challenging to recognize.

We have now specified the uncertainties by rephrasing the first sentence of the last paragraph to: “Despite the biases and uncertainties of the fossil record (e.g. selective preservation/sampling of rocks, animals, tissues, and environments),...”.

- This manuscript is unique because it emphasizes the utility of information derived from >500 bromalites from the Polish Basin. Yet, a synopsis of detailed information about the bromalite assemblages is difficult to find in the text. Figure 1 offers some information about how the bromalite assemblages compare with each other (although much of the type is unreadable and there is no explanation for numbers in the “Trophic structures” section of the figure).

Good comment, we have corrected it in the description of the figure 1. The numbers used in the trophic pyramids correspond to those presented in the faunal assemblages.

In addition, the supplemental material offers a detailed spreadsheet of the individual bromalites studied. Nevertheless, the paper would be much more informative if it included a table describing key characteristics of each bromalite assemblage. How many (percentages from each locality) bromalites contain specific dietary inclusions such as plants, arthropods, fish, or tetrapod parts? How many specimens in a given assemblage were studied with CT scans, thin sections, or surface analyses? What was the sedimentology/depositional environment of the site the assemblage was found in?

We planned to present these data in subsequent publications, which were to describe each bromalite assemblage in detail (sedimentology/taphonomy and geochemistry etc.). At the reviewer's suggestion, we are adding some of this information to the supplement of this manuscript (extra table describing key characteristics of each bromalite assemblage), we hope that it will fill these perceived gaps and allow the reader to better understand the context and character of the bromalite finds. We note here that more data will be presented in subsequent, more specialized publications.

How many of the bromalites in each assemblage are coprolites? This last question is relevant because gut contents provide different information from coprolites. Indeed, it is not clear from the text or supplements whether any of the bromalites from the Polish Basin appear to be gut contents or regurgitalites. (If all of the bromalites are coprolites, why not say this?) Since the manuscript stresses the unique approach of analyzing bromalites, I suggest that the authors lean into this innovative aspect of the paper by emphasizing the informative value of bromalites and providing comprehensive information about differences between the bromalite assemblages in a table or figure. This would help readers better understand how the bromalite assemblages changed through time and how these differences relate to changes in the vertebrate fauna.

Most of the bromalites are coprolites, followed regurgitalites and potentially some cololites. The broader term bromalite is used to include all of these.

In many cases, we were not sure about the identification of the specimens, whether they were coprolites or rather cololites, lithified contents of the digestive tract. It was especially difficult to determine in bone bed-type fossil deposits with various remains mixed. Here we were cautious and did not classify the specimens only as coprolites. In other words, in a situation of skeletal disarticulation, cololites may be incorrectly identified as coprolites.

Synopsis:

I thoroughly appreciate the integration of bromalite and body fossil evidence in documenting changes in trophic patterns from the Late Triassic to the Early Jurassic in this study; it is a refreshing approach to understanding the evolution of ancient dinosaur communities. However, I suggest that the authors strengthen the manuscript by: 1) directly relating the research results to the overarching question of why dinosaur taxa and populations increased at the same time that many non-dinosaurian vertebrates lost diversity or went extinct, 2) acknowledging that taphonomic factors often contribute to a biased fossil record of fossilized digestive residues, and 3) more strongly emphasizing the broader implications of this research.

We have strengthened these three areas by addressing the points raised by the reviewer (see above).

Miscellaneous questions:

Line 222:

What does the "one-sided content of Late Triassic herbivore bromalites" mean?

We have clarified this by rephrasing the sentence to "...the one-sided, conifer-dominated content of the Late Triassic herbivore bromalites."

Line 245:

Couldn't "reconfigurations of the floral assemblages in the region" have had a positive effect on some herbivores instead of having "knock-down effects on the tetrapod communities"?

This sentence has been rephrased to "...which in turn had large effects on the tetrapod communities." for clarification.

Reviewer Reports on the First Revision:

Referees' comments:

Referee #1 (Remarks to the Author):

I was impressed by the first draft of this manuscript, finding in particular the very original and through methodology and the high quality of the writing. The revised version is an improvement as the authors have worked diligently to incorporate the suggestions of the reviewers. I have no major issues with this work, but I do have some suggestions for minor changes that I think could result in additional improvement.

First, I appreciate the detailed responses of the authors to my comments and note that they have indeed included satisfactory additional comments on the paleoclimate of the Late Triassic, so I have no additional suggestions on this point. Also, I do look forward to seeing the additional work they have planned for a detailed paleoclimatic study coordinated with the present work

On my suggestion for providing additional global context to dinosaur radiation, the authors demur that this would be the next step in the study. I feel that just a little more context enhance the experience of the reader. This could be accomplished by simply adding a phrase here and there where specific taxonomic or ichnotaxonomic groups are introduced in the text. For example, when does *Stagonolepis* or forms similar to *Polonsuchus* first appear in other basins. Or when and where is the earliest appearances of *Grallator*, *Anchisaurupus*, *Kayentapus* and *Eubrontes*. This could be accomplished by adding a simple "... which first appeared during the early Norian in the ... basin."

Other minor points;

I. 76-78 How about a sentence that expands on the statement of the diachronous rise of dinosaurs by providing an example (e.g., global south vs north)?

L, 95, 113, 179 and a few other places, age is presented in very generalized terms in the text – oldest, younger, slightly younger, etc. I recognize that these ages are specified at various places the text and on figures 1 and 3, but presenting the specific age each time in the text helps the reader keep the chronology straight without having to refer to the figures.

Lastly, on the presentation of the pCO₂ levels in figure 1: As I suspected, the curve presented is derived from a single source based on $\delta^{13}\text{C}$ measurements of pedogenic carbonate . Estimates of pCO₂ based on pedogenic have the largest range of uncertainty; the pCO₂ estimate in Olsen et al. (2022) is based on measurements from the Newark Basin that differ strongly from measurements made in other basins (for example see Cleveland et al., 2008; <https://doi.org/10.1130/B26332.1>). More recent estimates of pCO₂ across the TJB based on stomatal indices are much better constrained, I suggest examining Roy et al. (2021; P3, Atmospheric CO₂ estimates based on Gondwanan (Indian) pedogenic carbonates reveal positive linkage with Mesozoic temperature variations) or Steinhorsdottir et al. (2011; P3, Extremely elevated CO₂ concentrations at the Triassic/Jurassic boundary) for a better perspective.

Referee #2 (Remarks to the Author):

Many thanks to the authors for their thoughtful and comprehensive responses to all reviewer comments. I have no further remarks and look forward to the publication of this manuscript.

Best wishes,
Emma Dunne

Referee #3 (Remarks to the Author):

Dear Editor,

I think the authors did an excellent job in integrating all the Reviewers' comments into their already quite detailed and extensively documented manuscript. I congratulate with their efforts and looking forward to seeing the final version of this study published in Nature.

I have spotted only a few minor faults to be fixed.

In Line 59, in text references 18-24 are coloured in red

Please invert the capitalisation of letters for Pco₂ (line 493), which should be uniformed to pCO₂ throughout the text

Best regards

Referee #4 (Remarks to the Author):

I appreciate the edits the authors have made to the manuscript; these revisions have improved the manuscript. However, I have a few more comments.

The addition of the phrase “Despite the biases and uncertainties of the fossil record (e.g. selective preservation/sampling of rocks, animals, tissues, and environments)...” to the last paragraph of the manuscript is a good reminder that “there is never such a thing as a complete picture of past ecosystems” (as written in the responses to the reviewers). Nevertheless, since this study bases its inferences on comparing ecosystems from different time intervals and localities, the potential impacts of taphonomic biases play a big role in the validity of the interpretations; this is especially true since conclusions are drawn from bromalites that have a lower preservation potential than skeletal materials. The inclusion of the new Supplementary Information Table 13 is very useful because it points out that: 1) the sedimentary environments from the localities from which bromalites have been recovered are interpreted as having been similar fluvial environments, and 2) usually rare herbivore bromalites have been recovered from all but the two oldest localities. Unfortunately, relatively few people read supplementary material, so if you do not mention these observations in the main text of the paper (if only briefly), in my view the conclusions are weakened by omitting evidence that supports the interpretations.

Other comments:

Supplementary Information Table 13

One of the headings in Table 13, “The ratio of phosphate/carbonate bromalites to those rich in plant matter” is confusing because both calcareous and (but less so) phosphatic coprolites can contain high concentrations of plant tissues. I suspect that the intent of this category is to compare the number of coprolites without significant plant tissues to those with substantial plant materials; if this is the case, I would describe it as such.

Figure 1:

This is a very intricate figure with lots of information, but I fear that readers will have as much trouble trying to read and understand these various graphics as I did. I hope the resolution of the published final figure will be higher because even when I viewed the graph at 400% on my screen I could not read the fine type (it was too blurry).

Figure 1; Trophic structures:

In the responses to the reviewers, the authors explained that “The numbers used in the trophic pyramids correspond to those presented in the faunal assemblages”; please expressly state this in the caption.

Figure 1; II. Morphotypes:

What do the different colors indicate for the gridded boxes -- are they simply unidentified (to readers) types of morphotypes? And what do the axes signify on these grids? It looks like diversity increases to the right, but what does the vertical axis signify?

Author Rebuttals to First Revision:

Firstly, we would like thank the four reviewers for their constructive feedback and valuable input to the manuscript!

Referees' comments:

Referee #1 (Remarks to the Author):

I was impressed by the first draft of this manuscript, finding in particular the very original and through methodology and the high quality of the writing. The revised version is an improvement as the authors have worked diligently to incorporate the suggestions of the reviewers. I have no major issues with this work, but I do have some suggestions for minor changes that I think could result in additional improvement. First, I appreciate the detailed responses of the authors to my comments and note that they have indeed included satisfactory additional comments on the paleoclimate of the Late Triassic, so I have no additional suggestions on this point. Also, I do look forward to seeing the additional work they have planned for a detailed paleoclimatic study coordinated with the present work. On my suggestion for providing additional global context to dinosaur radiation, the authors demur that this would be the next step in the study. I feel that just a little more context enhance the experience of the reader. This could be accomplished by simply adding a phrase here and there where specific taxonomic or ichnotaxonomic groups are introduced in the text. For example, when does *Stagonolepis* or forms similar to *Polonsuchus* first appear in other basins. Or when and where is the earliest appearances of *Grallator*, *Anchisauripus*, *Kayentapus* and *Eubrontes*. This could be accomplished by adding a simple "... which first appeared during the early Norian in the ... basin."

We agree with the reviewer; it would have been great to include more contextual data for the reader. Unfortunately, however, it is not possible for us to add sentences here and there due to the limited amount of space in *Nature*. We hope and believe that the manuscript (with support of figure 1&3 and Supp. Info.) still provides the reader with enough contextual information to get a robust picture of what our conclusions signify, also on a global scale.

Other minor points;

I. 76-78 How about a sentence that expands on the statement of the diachronous rise of dinosaurs by providing an example (e.g., global south vs north)?

This has been added.

L, 95, 113, 179 and a few other places, age is presented in very generalized terms in the text – oldest, younger, slightly younger, etc. I recognize that these ages are specified at various places the text and on figures 1 and 3, but presenting the specific age each time in the text helps the reader keep the chronology straight without having to refer to the figures.

This has been specified in all the mentioned places as well as on line 141 ("post-Carnian").

Lastly, on the presentation of the pCO₂ levels in figure 1: As I suspected, the curve presented is derived from a single source based on δ¹³C measurements of pedogenic carbonate. Estimates of pCO₂ based on pedogenic have the largest range of uncertainty; the pCO₂ estimate in Olsen et al. (2022) is based on measurements from the Newark Basin that differ strongly from measurements made in other basins (for example see Cleveland et

al., 2008; <https://doi.org/10.1130/B26332.1>). More recent estimates of pCO₂ across the TJB based on stomatal indices are much better constrained, I suggest examining Roy et al. (2021; P3, Atmospheric CO₂ estimates based on Gondwanan (Indian) pedogenic carbonates reveal positive linkage with Mesozoic temperature variations) or Steinthorsdottir et al. (2011; P3, Extremely elevated CO₂ concentrations at the Triassic/Jurassic boundary) for a better perspective.

Thanks for bringing the results and problems with determining CO₂ concentrations in the geological record to our attention.

Our comment here will rather be the statement of non-experts, for the purposes of this publication we did not perform any geochemical studies of palaeosoils or stomatal index-based estimates of pCO₂. However, to illustrate some global processes (e.g. changes in concentration of atmospheric CO₂), which influenced the evolution of terrestrial ecosystems, we used information, i.e. the pCO₂ plot, presented in other publications. In this case data from Olsen et al. (2022: *PNAS*), which were presented earlier in Schaller et al. (2012; *Geol. Soc. Am. Bull.*) and Schaller et al. (2015; *Earth Planet. Sci. Lett.*). It is difficult for us to evaluate the results presented in Schaller et al. (2012, 2015) and Olsen et al. (2022). The publication by Roy et al. (2021) presents the results for atmospheric CO₂ concentration for the late Early and Middle Triassic. There are no new data for the Late Triassic or earliest Jurassic in this study. Roy et al. (2021: figs. 6 and 9) collate data from other publications and compare results based on studies of the pedogenic record with those obtained from stomatal indices. The publication includes a commentary on these.

“The extremely high pCO₂ values (>4000 ppmV) obtained for the majority of the Late Triassic do not seem to be accompanied by any major changes in temperature (Fig. 9; Schaller et al., 2012, Schaller et al., 2015; Knobbe and Schaller, 2018). Since the temperature anomaly record was estimated relative to the average temperature value of the Late Triassic (~23.9 °C), the relative temperature changes were probably muted for this period (Schaller et al., 2015; Knobbe and Schaller, 2018). The soil-depth-based S(z) value of ~3000 ppmV assumed in the study is also on the higher side, which may have further resulted in overestimation in pCO₂ values, and contributed to the apparent decoupling observed between pCO₂ and temperature values.”

The results presented by Steinthorsdottir et al. (2011) indicate that pre-TJB (late Rhaetian of Jameson Land, Greenland and Larne, Ireland), the CO₂ concentration was approximately 1000 ppm, that it started to rise steeply pre-boundary and had doubled to around 2000–2500 ppm at the TJB. These are data from a very small part of the Triassic and Jurassic covering the end of the Rhaetian and the beginning of the Hettangian, a rather short time period. They do not differ substantially from the results for the TJB interval presented in Olsen et al. (2022; range from 1000 to 3000 ppm, but there are also major peaks of 4000 ppm and even exceeding 5000 ppm).

Without entering into further discussion on the validity of one method or the other, our intention was to show data on atmospheric carbon dioxide concentration covering a significant part of the Late Triassic (Carnian-Rhaetian interval) and the earliest Jurassic. We also reached for data that were obtained from a continental sequence that is well studied and stratigraphically well defined.

Referee #2 (Remarks to the Author):

Many thanks to the authors for their thoughtful and comprehensive responses to all reviewer comments. I have no further remarks and look forward to the publication of this manuscript.

Best wishes,
Emma Dunne

Referee #3 (Remarks to the Author):

Dear Editor,

I think the authors did an excellent job in integrating all the Reviewers' comments into their already quite detailed and extensively documented manuscript. I congratulate with their efforts and looking forward to seeing the final version of this study published in Nature.

I have spotted only a few minor faults to be fixed.

In Line 59, in text references 18-24 are coloured in red

This has been fixed (and reference “24” has moved to its correct place in the reference list).

Please invert the capitalisation of letters for Pco2 (line 493), which should be uniformed to pCO2 throughout the text

Corrected.

Best regards

Referee #4 (Remarks to the Author):

I appreciate the edits the authors have made to the manuscript; these revisions have improved the manuscript. However, I have a few more comments.

The addition of the phrase “Despite the biases and uncertainties of the fossil record (e.g. selective preservation/sampling of rocks, animals, tissues, and environments)...” to the last paragraph of the manuscript is a good reminder that “there is never such a thing as a complete picture of past ecosystems” (as written in the responses to the reviewers). Nevertheless, since this study bases its inferences on comparing ecosystems from different time intervals and localities, the potential impacts of taphonomic biases play a big role in the validity of the interpretations; this is especially true since conclusions are drawn from bromalites that have a lower preservation potential than skeletal materials.

The inclusion of the new Supplementary Information Table 13 is very useful because it points out that: 1) the sedimentary environments from the localities from which bromalites have been recovered are interpreted as having been similar fluvial environments, and 2) usually rare herbivore bromalites have been recovered from all but the two oldest localities.

Unfortunately, relatively few people read supplementary material, so if you do not mention these observations in the main text of the paper (if only briefly), in my view the conclusions are weakened by omitting evidence that supports the interpretations.

We agree with the reviewer on these points and the mentioned observations from Supp. Table 13 are added to the last paragraph of the manuscript.

Other comments:

Supplementary Information Table 13

One of the headings in Table 13, “The ratio of phosphate/carbonate bromalites to those rich in plant matter” is confusing because both calcareous and (but less so) phosphatic coprolites can contain high concentrations of plant tissues. I suspect that the intent of this category is to compare the number of coprolites without significant plant tissues to those with substantial plant materials; if this is the case, I would describe it as such.

This is now clarified in the heading.

Figure 1:

This is a very intricate figure with lots of information, but I fear that readers will have as much trouble trying to read and understand these various graphics as I did. I hope the resolution of the published final figure will be higher because even when I viewed the graph at 400% on my screen I could not read the fine type (it was too blurry).

Good point, the figure has been slightly modified to meet the editorial requirements, it is planned to be printed in a width of 180 mm (two columns), where the smallest font will be 5 pt. The editorial office will receive a high-resolution image (600 dpi, the previous one was only 300 dpi) and the original CDR file, which will be able to be exported with an even higher resolution.

Figure 1; Trophic structures:

In the responses to the reviewers, the authors explained that “The numbers used in the trophic pyramids correspond to those presented in the faunal assemblages”; please expressly state this in the caption.

This sentence was moved before the sentence starting with “Vertebrate bromalites:” in the caption, and was introduced by “Trophic structures:” for clarification.

Figure 1; II. Morphotypes:

What do the different colors indicate for the gridded boxes -- are they simply unidentified (to readers) types of morphotypes? And what do the axes signify on these grids? It looks like diversity increases to the right, but what does the vertical axis signify?

The colors indicate the morphotypes of bromalites, a way of illustrating the diversity of morphotypes found in individual faunal communities. The colors therefore do not encode the same morphotype for all assemblages. Coloring the individual morphotypes from each community is more readable in figure like this than using terms such as morphotype “a, b c” or “1, 2, 3”. Importantly, this graphic shows the diversity in individual communities, but is not information about the overall diversity for the entire Triassic-Jurassic sequence. More data on morphotypes and their diversity in specific communities is provided in the supplementary tables.